# Gemistocytic tumor cells programmed for glial scarring characterize T cell confinement in IDH-mutant astrocytoma

Levi van Hijfte [1,2] ✉, Marjolein Geurts[1,2], Iris de Heer[1], Santoesha A. Ghisai [1], Hayri E. Balcioglu [2], Youri Hoogstrate [1], Wies R. Vallentgoed [1], Rania Head[1], Rosa Luning[1], Thierry van den Bosch[3], Bart Westerman [4], Pieter Wesseling [5,6], Johanna A. Joyce [7,8,9], Pim French [1,10] ✉ & Reno Debets [2,10] ✉

Isocitrate dehydrogenase 1/2 mutant (IDHmt) astrocytoma is considered a T cell-deprived tumor, yet little is known regarding the phenotypes underlying T cell exclusion. Using bulk, single nucleus and spatial RNA and protein profiling, we demonstrate that a distinct spatial organization underlies T cell confinement to the perivascular space (T cell cuff) in IDHmt astrocytoma. T cell cuffs are uniquely characterized by a high abundance of gemistocytic tumor cells (GTC) in the surrounding stroma. Integrative analysis shows that GTC-high tumors are enriched for lymphocytes and tumor associated macrophages (TAM) and express immune cell migration and activation programs. Specifically, GTCs constitute a distinct sub-cluster of the astrocyte-like tumor cell state that co-localizes with immune reactive TAMs. Neighboring GTCs and TAMs express receptor-ligand pairs characteristic of reactive astrogliosis and glial scarring, such as SPP1/CD44 and IL-1β/IL1R1. Collectively, we reveal that T cell confinement in IDHmt astrocytomas associates with GTC-TAM networks that mimic glial scarring mechanisms.

Efforts to treat glioma with immune therapies, such as checkpoint blockade, have thus far shown no clinical benefit[1–3]. The presence of intra-tumoral CD8 + T cells is considered a robust predictor for the success of immune therapies and is generally low in glioma[4–6]. The low abundance of CD8 + T cells has been attributed to factors like low tumor mutational burden and the inherently immune suppressive properties specific to the central nervous system (CNS)[7–11]. In isocitrate dehydrogenase 1/2 mutated (IDHmt) astrocytomas, the oncometabolite D-2-hydroxyglutarate has been reported to further inhibit CD8 + T cell effector function directly and limit the expression of chemo-

attractants CXCL9 and CXCL10[12–14]. Several of these mechanisms are still debated and further research is required to form an integrated view of the immune tumor microenvironment (TME) in glioma[10,11].

Gliomas have also been shown to actively adopt pathways that can regulate immune cell infiltration such as the recruitment and reprogramming of tumor associated tissue resident microglia and monocyte-derived macrophages (collectively named tumor associated macrophages (TAM)) to inhibit T cell function and aid tumor growth[5,15–18]. The underlying pathways and interactions responsible for this within the complex cellular networks in glioblastoma are being

[1]Department of Neurology, Erasmus MC Cancer Institute, Rotterdam, The Netherlands. [2]Department of Medical Oncology, Laboratory of Tumor Immunology, Erasmus MC Cancer Institute, Rotterdam, The Netherlands. [3]Department of Pathology, Erasmus Medical Center, Rotterdam, The Netherlands. [4]Department of Neurosurgery, Amsterdam UMC/VUMC, Amsterdam, The Netherlands. [5]Department of Pathology, Amsterdam UMC/VUMC and Brain Tumour Center, Amsterdam, The Netherlands. [6]Princess Máxima Center for Pediatric Oncology, Utrecht, The Netherlands. [7]Department of Oncology, University of Lausanne, Lausanne, Switzerland. [8]Ludwig Institute for Cancer Research, Lausanne, Switzerland. [9]Agora Cancer Center Lausanne and Swiss Cancer Center Léman, Lausanne, Switzerland. [10]These authors jointly supervised this work: Pim French, Reno Debets. ✉e-mail: l.vanhijfte@erasmusmc.nl; p.french@erasmusmc.nl; j.debets@erasmusmc.nl

unraveled in single-cell studies which facilitate the design of more appropriate immune therapies[19–21]. Spatially resolved investigations consequently highlighted that relationships between functional immune cell states and tumor cell subtypes in glioblastoma can be a local occurrence, showing specific interactions between transcriptionally distinct tumor cells and TAMs that can aid T cell exhaustion[22–24]. Thus far, such studies are lacking for IDHmt astrocytoma and even though its TME is considered to be mostly immune-suppressive with little T cell infiltration, it remains challenging to consistently classify all functional immune cell states due to their plasticity, which can rely greatly on their spatial embedding in the TME and could be essential for appropriate design of novel therapies[5,16,25,26].

Here, we aim to characterize the spatial composition of the immune TME in IDHmt astrocytoma. Using a combination of bulk, single nucleus, and spatial transcriptomics alongside spatial proteomics techniques and histology on a large cohort of clinically well-defined IDHmt astrocytomas, we identify an immune phenotype that underlies T cell exclusion. We demonstrate that T cell confinement to the perivascular space is characterized by co-localization of distinct tumor cells called gemistocytic tumor cells (GTC) and reactive TAMs in the surrounding stroma. Together, GTCs and reactive microglia highlighted pathways associated with glial scarring, a CNS-specific mechanism of scar formation seen in CNS pathologies that is involved in the attenuation of immune cell invasion.

## Results

### T cells accumulate in the perivascular space in IDH-mutant astrocytoma

We profiled the immune compartment in the IDHmt astrocytoma microenvironment with a multimodal approach using paired primary and recurrent resection samples from a large patient cohort (Fig. 1a, b; Supplementary Fig. 1a). Using multiplex immunofluorescence (IF) stainings, 75 paired primary and recurrent tumor samples from 40 patients were stained for T cells, TAMs, B cells and tumor cells (GLASS-NL cohort[27]; characteristics can be found in Supplementary Data 1). Of all tumor samples, 42 were classified as CNS WHO grade 2, 11 as grade 3 and 22 as grade 4. Overall, 1292 images (median of 16 images per sample) were collected from tumor cell high- and low regions of interest (ROI; Supplementary Fig. 1b–d). Even though there was considerable inter- and intra-tumoral heterogeneity (Supplementary Fig. 1e, f), we observed an overall scarcity of lymphocytes, a predominance of TAMs in tumor-high regions and otherwise indiscriminate quantities of immune cells regarding tumor density, grade or resection number (Supplementary Fig. 1g–i).

Even though their overall counts were low, a comparison across all ROIs showed that an increase in T cell count coincided with clustering of lymphocyte subsets that typically occurred around blood vessels (based on z-scores, negative and positive values represent lower and higher inter-cell distances than predicted by chance distribution for the number of cells in that particular field, see methods; Fig. 1c, d). In inflamed conditions, leukocytes can accumulate around blood vessels in the CNS perivascular space, which is delineated by the endothelial and parenchymal basement membranes[28]. To determine whether vessel-adjacent T cells reside in the perivascular space, we stained 15 IDHmt astrocytoma samples containing high T cell quantities for CD3, CD31, and Laminin-α2 (LAMA2; a marker for the parenchymal basement membrane). T cells showed a significantly smaller distance to CD31+ vascular structures compared to other cells (median distance of 21 µm and 44 µm; Kolmogorov–Smirnov test; D = 0.24441, $p < 2.2 \times 10^{-16}$; Fig. 1e, f, Supplementary Fig. 2a, b). Importantly, T cells accumulated within LAMA2 demarcated regions in all samples, indicating that T cells reside in the perivascular space behind the parenchymal basement membrane in these tumors (Fig. 1e, g; Supplementary Fig. 2a, c).

In addition to perivascular confinement of T cells (T cell cuff), ROIs showed T cell presence in the stroma and, in most cases, T cells were absent (15.2%, 29.7% and 55.3% respectively; Fig. 1d). Most samples showed all three T cell related tissue phenotypes without a statistically significant difference between initial or recurrent tumors (chi-square test, $p = 0.073$; Supplementary Fig. 2d). Lymphocytes were more abundant in T cell cuffs than in stroma, independently of tumor cell numbers (Fig. 1h, Supplementary Fig. 2e). There was no difference in TAM quantity between ROIs with T cell cuffs or stromal T cells, but TAM numbers were significantly lower in T cell absent ROIs. As expected, inter-cell distance scores among B cells, CD8 T cells and CD4 T cells were lowest in T cell cuff ROIs (Fig. 1i). Conversely, the inter-cell distance z-score for TAMs to lymphocyte subsets was significantly higher in T cell cuff ROIs, indicating that perivascular lymphocyte clustering does not involve TAMs.

T cell cuffs were present in most samples and the fraction of samples containing one or more T cell cuff was independent of WHO grade or initial/recurrent resection (Supplementary Fig. 3a). Relative T cell cuff ROI quantities (rather than complete tumor sample) were similarly independent of resection (Supplementary Fig. 3b–d). All leukocyte quantities, but not tumor cell quantities, increased with cuff size (Supplementary Fig. 3e). Lymphocyte subset clustering increased significantly together with cuff size, whereas the inverse was seen for TAM clustering with lymphocytes (Supplementary Fig. 3f). Remarkably, within the T cell population, the fraction of CD8 T cells increased significantly with an increase in cuff size (Supplementary Fig. 3g). These results recapitulated known patterns of low overall lymphocyte counts and we characterized conserved T cell related tissue phenotypes in IDHmt astrocytoma[5].

### GTCs accumulate around T cell cuffs and associate with immune cell recruitment and activation

Histological evaluation of ROIs containing T cell cuffs revealed accumulation of gemistocytic cells in the surrounding tumor stroma that were positive for the IDH1-R132H antigen, demonstrating their neoplastic origin (Fig. 2a). GTCs are histologically distinct due to their large cell body with a uniformly stained eosinophilic cytoplasm and eccentrically placed nucleus. The 2021 WHO classification of CNS tumors (WHO 2021) assigns the label "gemistocytic astrocytoma" when >20% of malignant cells show gemistocyte morphology. Due to unclear consequences of the presence of GTCs in these tumors, attention for this tumor subtype has waned. Electron microscopy (EM) images of GTCs showed densely stacked filaments that were absent in other, non-gemistocytic cells, which is in concordance with other EM reports of these cells in glioma (Fig. 2b, Supplementary Fig. 4)[29]. Importantly, GTCs greatly resemble reactive astrocytes present in other CNS pathology like acute multiple sclerosis (MS) lesions[30]. Of note, the voluminous cytoplasm and the regular occurrence of multiple, eccentrically placed nuclei seen in GTCs could alternatively be explained by cell engulfment by means of phagocytosis or entosis, processes that have been described in reactive astrocytes and tumor cells[31,32]. However, we did not find any clear evidence of these phenomenon in our EM recordings.

All immune cell counts increased significantly and incrementally with the GTC quantity independent of T cell related tissue phenotypes (Fig. 2c; Supplementary Fig. 5a, b). ROI GTC scores were highly heterogeneous within tumor samples and their abundance often changed at tumor recurrence (Supplementary Fig. 5c). T cell cuff ROIs showed a significantly higher abundance of GTCs compared to other T cell related spatial phenotypes, even though GTCs were present within all ROI annotation groups, and quantity and size of T cell cuffs positively correlated with GTC quantity (chi-square test; $p < 0.01$ for all tests between the three tissue phenotypes, Fig. 2d; chi-square test;

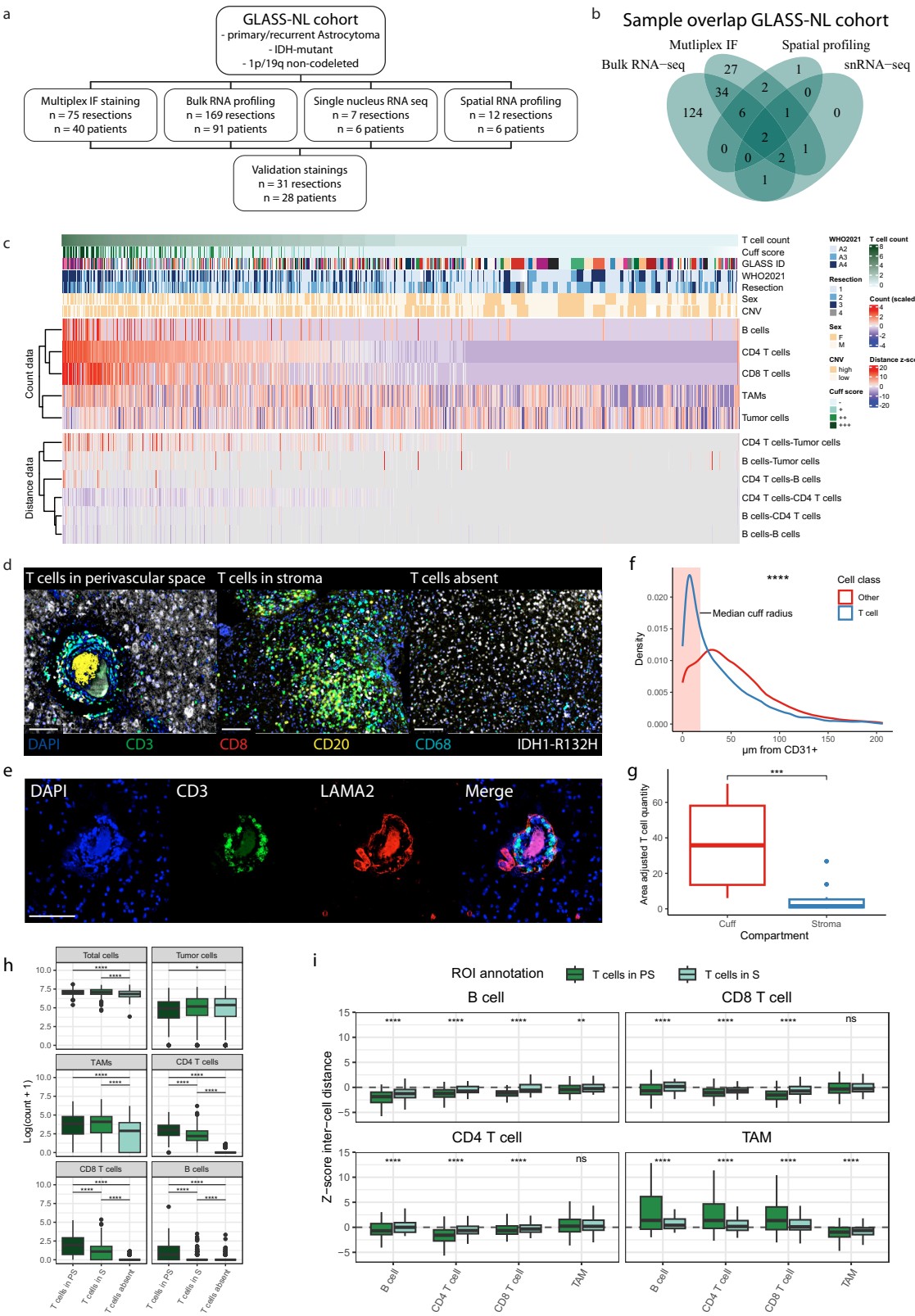

$p < 2.2 \times 10^{-16}$; Fig. 2e). It is noteworthy that GTC presence was slightly but significantly higher in initial tumors compared to their recurrences and WHO grade 3 samples showed the highest abundance of GTCs (Chi-square test; $p = 0.00294$, $p = 8.80 \times 10^{-11}$, respectively; Supplementary Fig. 5c–e). At sample level, comparisons of average GTC scores with cell quantities showed a correlation with immune cell, but not tumor cell quantities (Supplementary Fig. 5f).

To further explore the potential relationship between GTCs and immune cell populations, we transcriptomically characterized 157 IDHmt astrocytomas from the GLASS-NL cohort that were split in two classes based on histological assessment of GTC abundance according to the thresholds set by WHO 2021 (cutoff of 20% GTCs in malignant cell population; Supplementary Data 2). The differential gene expression (DGE) test resulted in 1312 upregulated genes in GTC-high ($n = 39$)

**Fig. 1 | T cell distributions show distinct spatial tissue phenotypes in IDHmt astrocytoma. a** Flowchart of study setup and methods. **b** Venn diagram of the sample n for each analysis modality. **c** Heatmap of log-transformed cell counts and inter-cell distance z-scores ($n = 1293$ ROIs; $n = 75$ samples). Columns are ordered by ROI T cell count, top annotations show patient and ROI characteristics. For distance data, blue indicates a negative z-score and higher cell clustering than expected. Red indicates a positive z-score and higher cell dispersion than expected.
**d** Representative images from multiplex IF stainings of three tissue phenotypes based on T cell quantity and location; T cells in the perivascular space (T cell cuff), T cells in the stroma and T cells absent regions. **e** Representative IF image of CD3 (T cells) and LAMA2 (parenchymal basement membrane) for a T cell cuff. **f** Cell distance to nearest CD31+ vessel structure. The red area indicates the median cuff radius of T cell cuffs containing three or more T cells. Kolmogorov–Smirnov test, two-sided ($n = 11$; $p < 2.2 \times 10^{-16}$). **g** Area-adjusted T cell counts for cuff area and stromal area. Wilcoxon rank sum test, two-sided ($n = 13$; $p = 0.0002$). **h** Count values for all cell subsets separated for ROI tissue phenotype ($n = 667$ T cells absent,

$n = 184$ T cells PS, $n = 359$ T cells in S). Wilcoxon rank sum test, two-sided, fdr corrected ($p$ values from top to bottom per header: Total cells: $2.57 \times 10^{-8}$, $3.38 \times 10^{-15}$; Tumor cells: 0.021; TAMs: $3.55 \times 10^{-10}$, $1.23 \times 10^{-21}$; CD4 T cells: $5.09 \times 10^{-122}$, $1.62 \times 10^{-12}$, $2.90 \times 10^{-167}$; CD8 T cells: $2.44 \times 10^{-104}$, $5.13 \times 10^{-12}$, $1.99 \times 10^{-89}$; B cells: $1.54 \times 10^{-51}$, $9.41 \times 10^{-21}$, $3.68 \times 10^{-7}$). **i** Inter-cell distance z-scores for immune cell subsets in tissue phenotypes. Boxplots display distance between cell types indicated in the plot title (from) and x-axis annotations (to). Wilcoxon rank sum test, two-sided, fdr corrected (n for all ROI groups are shown in Supplementary Data 15; p values from left to right per header: B cell: $2.53 \times 10^{-10}$, $8.81 \times 10^{-25}$, $4.68 \times 10^{-25}$, 0.0011; CD8 T cell: $2.50 \times 10^{-9}$, $1.23 \times 10^{-24}$, $2.98 \times 10^{-50}$, 0.0972; CD4 T cell: $2.21 \times 10^{-5}$, $2.40 \times 10^{-78}$, $2.19 \times 10^{-11}$, 0.1477; TAM: $7.55 \times 10^{-5}$, $9.29 \times 10^{-39}$, $8.11 \times 10^{-16}$, $1.96 \times 10^{-6}$). Scale bars in (**d**, **e**) show 100 μm. S stroma, PS perivascular space, TAM tumor associated macrophage, ROI region of interest, CNV copy number variation, ns not significant. Boxplots in (**g–i**) show the hinges at the first and third quartiles with the median as the center. The whiskers show min and max value until 1.5 times the interquartile range.

and 1642 upregulated genes in GTC-low ($n = 118$) tumors ($p < 0.05$, |log2FC| > 0.5, fdr adjusted; Fig. 2f). Many of the upregulated genes in GTC-high tumors were representative of T cells and TAMs. Pathway analysis highlighted the occurrence of extravasation, immune activation and antigen presentation in GTC-high versus GTC-low tumors (Fig. 2g). Notably, the PD-1/PD-L1 immune checkpoint pathway was the only immune-related pathway that was downregulated in GTC-high tumors.

Next, we tested whether spatial gene expression results recapitulated the patterns found in bulk RNA-seq. To this end, we used expression of 1825 immune related genes to profile 72 ROIs collected from 12 tumors on the NanoString GeoMx Digital Spatial Profiler (DSP; methods; Supplementary Data 3)[33]. DGE analysis of GTC-high versus GTC-low ROIs yielded 42 upregulated and 45 downregulated genes ($p < 0.05$, |log2FC| > 0.5, fdr adjusted; Supplementary Fig. 5g). 27 of the upregulated and 2 or the downregulated genes were TAM-related markers. No T cell related genes were found as T cell cuffs were excluded from ROI selection. Importantly, a direct comparison of the 1515 overlapping genes between the spatial and bulk DGE analyses showed a high agreement (Supplementary Fig. 5h). These results show that T cell cuffs, and immune cell quantities in general, are closely related to the presence of GTCs in IDHmt astrocytoma, which is substantiated by the observed upregulated gene expression of immune cell activation pathways in GTC-high tumors.

### GTCs represent a distinct cell subpopulation that is characterized by glial scarring

To determine which cell types are responsible for immune activation pathways in GTC-high tumors, we performed single nucleus RNA-sequencing (snRNA-seq) on 7 cryopreserved IDHmt astrocytoma samples from the GLASS-NL cohort (Fig. 3a; Supplementary Fig. 5i; Supplementary Data 4). We achieved satisfactory data complexity that is comparable to single cell RNA sequencing data from fresh tissue regardless tissue age, enabling identification of all major glioma specific cell types (see methods for details; Supplementary Fig. 6a–d). We observed high concordance between copy number variation estimates of snRNA-seq tumor cell clusters and genome-wide DNA methylation profiles (Supplementary Fig. 6e–g). Even though we selected for a specific neoplastic cell population, the majority of differentially expressed (DE) genes from GTC-high tumors were expressed by TAMs and T cells, with a subset that was highly expressed by pericytes/endothelial cells (Supplementary Fig. 6h). Because the expression of GTC-high DE genes could be restricted to an unrecognized subpopulation, we analyzed tumor cells separately and identified 17 clusters that recapitulated known transcriptional IDHmt astrocytoma tumor cell states (i.e., stem-like, oligo-like and astrocyte-like; Fig. 3b; Supplementary Fig. 6c)[34]. The bulk GTC-high DE genes were mostly expressed in astrocyte-like tumor cell subpopulations (Fig. 3c). Indeed,

the corresponding enrichment scores for bulk GTC DE genes highlighted a distinct cluster of cells within the astrocyte-like tumor cell state that was derived from all 7 tissue samples (Fig. 3d; Supplementary Fig. 6i, j).

We noted that GTCs show histological resemblance to reactive astrocytes, a cell type that is involved in glial scar formation in various pathologies[35–39]. To determine whether this resemblance is extended in their transcriptomic profile, we compiled a reactive astrocyte marker gene list from literature (Supplementary Data 11)[40–45]. These marker genes showed increased log2fc in bulk GTC-high samples and showed increased expression in the newly identified snRNA-seq GTC cell cluster (Fig. 3e, f). To confirm this on protein level, we characterized presence of 54 proteins in 72 ROIs from tissue sections consecutive to the ones used for spatial transcriptomics on the NanoString GeoMx DSP (Supplementary Fig. 7a, b; Supplementary Data 5). Of the analyzed proteins, only OX40L, CD127, PTEN, and CD44 were significantly upregulated in GTC-high ROIs (Fig. 3g). OX40L and CD127 are generally associated with T cell function, whereas PTEN is associated with cell cycle control. CD44 is a ubiquitously expressed molecule that is extensively described as a marker for IDHwt and IDHmt glioma as well as for reactive astrogliosis. These results demonstrate that GTCs form a distinct transcriptional tumor cell subpopulation that express markers for glial scarring.

### GTCs and reactive TAMs form a cellular network that contributes to a glial scarring-like program

To examine the interactions between GTCs and immune cell populations in a spatial context, we used our NanoString GeoMx DSP spatial transcriptomics data with samples that were annotated for the three T cell related tissue phenotypes (Fig. 4a; Supplementary Fig. 7c). We performed weighted gene co-expression network analysis (WGCNA) and identified 13 gene modules that were often specific for cell types identified in our snRNA-seq data and showed distinct correlation patterns between ROI traits (Supplementary Fig. 7d, e). For example, gene module 9, containing the genes *TOP2A* and *MCM4*, correlated with resection/tumor grade and was mainly expressed by G1/S/G2/M tumor cells, providing support for the existing notion that higher-grade tumors contain larger proportions of proliferating malignant cells[27]. Gene modules 3, 4 and 5 were positively correlated with the T cell cuff annotation, and the association scores of all genes consequently showed a high correlation between GTC and T cell cuff annotations with positive scores for genes from modules 3, 4, and 5 in both annotations (Fig. 4b, c). A large proportion of genes from these three modules were expressed by TAMs in our snRNA-seq data, confirming our observation that T cell cuffs are positively correlated with TAM quantities (Fig. 4d). While gene modules 2 and 5 were uniformly expressed across all TAMs, gene module 3 was only expressed by a subset with a uniform microglia transcriptomic profile (Fig. 4e–g;

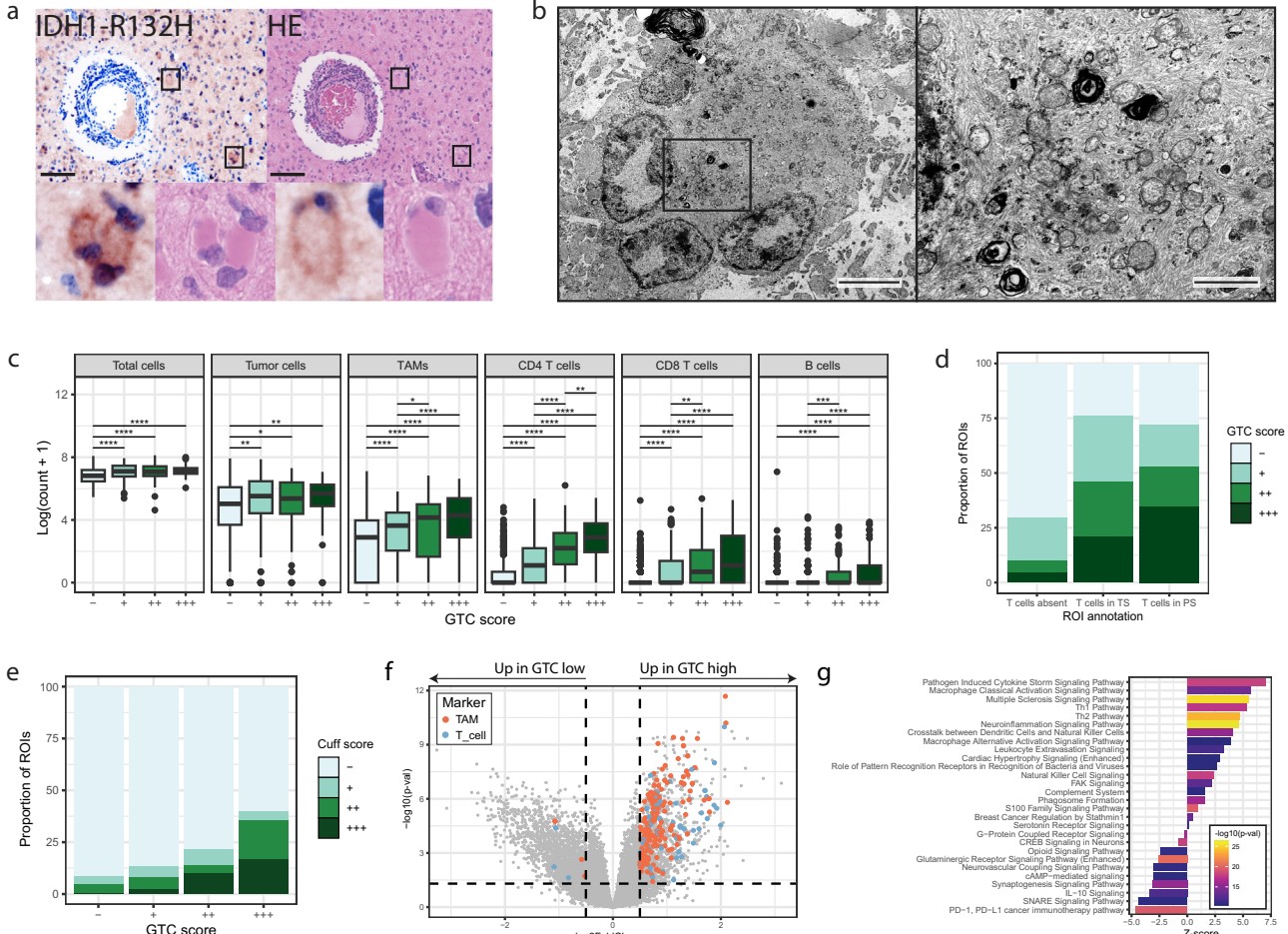

**Fig. 2 | Gemistocytic tumor cells are localized in tumor stroma surrounding T cell cuffs and associate with the accumulation of immune cells.**
**a** Representative images of IDH1-R132H and H&E staining of the stroma surrounding a T cell cuff. Insets show gemistocytes that stain positive for the IDH1-R132H mutation. **b** Electron microscopy images of a gemistocytic tumor cell (GTC; left, magnification = 6000) and a magnification of the insert (right, magnification = 25.000). **c** ROI (immune) cell quantities separated by GTC scores ($n = 494$ −, $n = 216$ +, $n = 122$ ++, $n = 127$ +++). Wilcoxon rank sum test, two-sided, fdr corrected ($p$ values from top to bottom per header: Total cells: $1.16 \times 10^{-10}$, $5.57 \times 10^{-7}$, $1.16 \times 10^{-10}$; Tumor cells: $0.0003$, $0.0115$, $0.0011$; TAMs: $0.0193$, $4.30 \times 10^{-5}$, $4.00 \times 10^{-12}$, $2.22 \times 10^{-7}$, $2.46 \times 10^{-5}$; CD4 T cells: $0.0046$, $9.90 \times 10^{-7}$, $3.63 \times 10^{-15}$, $1.80 \times 10^{-42}$, $1.09 \times 10^{-28}$, $1.09 \times 10^{-17}$; CD8 T cells: $0.006$, $6.86 \times 10^{-6}$, $2.38 \times 10^{-22}$, $2.11 \times 10^{-16}$, $8.73 \times 10^{-10}$; B cells: $9.00 \times 10^{-5}$, $2.26 \times 10^{-7}$, $4.69 \times 10^{-16}$, $1.16 \times 10^{-10}$). Four bins were used to separate GTC quantity (Supplementary Fig. 5a). **d** Fractions of GTC ROI scores for the three spatial T cell phenotypes ($n = 560$ T cells absent,

$n = 144$ T cells PS, $n = 255$ T cells in S). **e** Faction T cell cuff size for GTC scores ($n = 494$ −, $n = 216$ +, $n = 122$ ++, $n = 127$ +++). **f** Volcano plot of the differential gene expression test between GTC-high ($n = 39$; positive log2fc) and GTC-low ($n = 118$; negative log2fc) tumors from bulk RNA sequencing. Colors indicate marker genes for T cells and TAMs. Wald test, two-sided, fdr corrected. Lines indicate fdr-adjusted $p$-value and log2FoldChange cutoffs. **g** Gene set enrichment analysis for the differentially expressed genes from (**f**) between GTC-low (negative z-score) and GTC-high (positive z-score) samples. Fisher's Exact Test, right-tailed. Scale bars in (**a**) show 100 μm. Scale bars in (**b**) show 4 μm (left) and 1 μm (right). S stroma, PS perivascular space, TAM tumor associated macrophage, GTC gemistocytic tumor cell, ROI region of interest. All GTC quantities were evaluated by three independent reviewers. Boxplots in (**c**) show the hinges at the first and third quartiles with the median as the center. The whiskers show min and max value until 1.5 times the interquartile range.

Supplementary Fig. 7f, g). Module 3 positive TAMs upregulated several chemokines (*CCL3*, *CCL4L2* and *CCL4*) and markers for reactive microglia (*SRGN*, *CD83*, *SPP1*, *FOS* and *IL1B*; Fig. 4h). Importantly, module 3 TAMs were also unique in the expression of signaling molecules that are involved in reactive astrogliosis (*SPP1*, *IL1B*, *C1QA*, *C1QB* and *TNF*, Fig. 4i). Next, we assessed whether GTCs and module 3 TAMs could directly interact by inferring cell-cell interaction pathways on our snRNA-seq data and found 54 significant pathways (Supplementary Fig. 7h). One of the top identified pathways was SPP1-CD44 signaling, which is involved in a wide variety of functions, among which glial scarring, and module 3 TAMs and GTCs were identified as likely senders and receivers, respectively (Fig. 4j). Notably, a comparison between the top DE genes for GTCs and module 3 TAMs derived from snRNA-seq data (log2fc > 1 and adjusted $p < 0.001$, Supplementary Data 6) showed high correlations in both bulk and spatial RNA-seq data

($\rho = 0.55$, $p < 2.2 \times 10^{-16}$; $\rho = 0.72$, $p < 2.2 \times 10^{-16}$; Fig. 5a, b), showing that the association between GTCs and module 3 TAMs is recapitulated across our data sets.

We performed multiplex IF stainings guided by transcriptomic markers to visualize the relationship between GTCs, TAMs and T cells (Supplementary Data 7; Supplementary Fig. 8a–e). Since no formal marker for GTCs exists, we used CD44 and CRYAB to identify GTCs as these two markers were the most conserved across our datasets. Indeed, in our IF stainings, both CD44 and CRYAB specifically stained GTCs in both GTC-high and GTC-low tumor samples and the number of CD44+ or CRYAB+ cells were significantly higher in GTC-high tumors (Wilcoxon rank sum test, two-sided; $p = 0.00666$, $p = 0.00524$, respectively; Fig. 5c–f). Whole slide nearest neighbor (NN) analysis showed consistent clustering within a cell type (GTCs, TAMs, and T cells) independent of GTC abundance, but not between

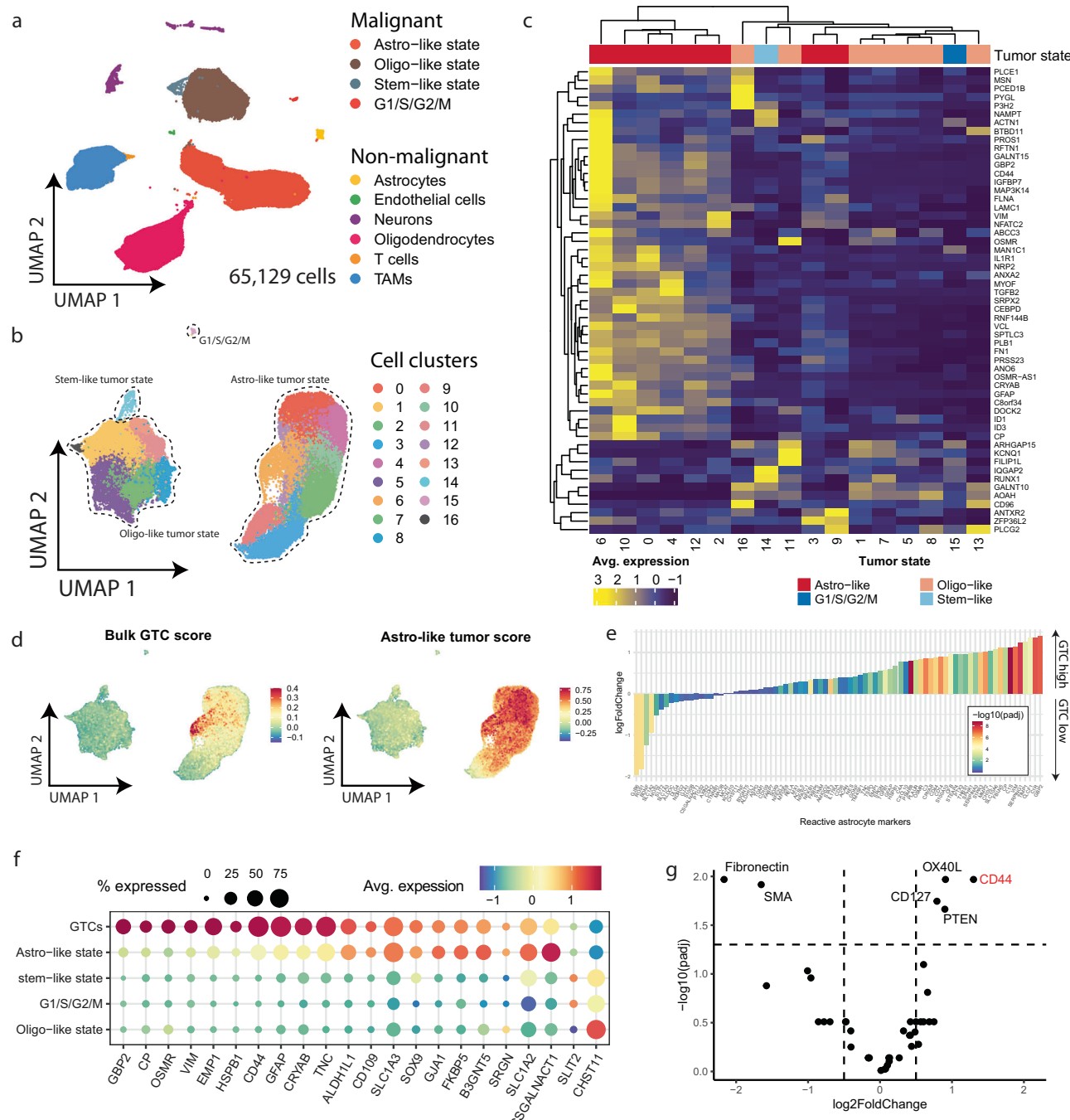

**Fig. 3 | Gemistocytic tumor cells represent a transcriptionally distinct subpopulation that expresses reactive astrogliosis markers.** UMAP of all cells (**a**; n = 65,129) and tumor cells (**b**; n = 42,011) for integrated snRNA-seq glioma samples (n = 7). Estimated transcriptional tumor cell states from Supplementary Fig. 6c are indicated. **c** Expression of bulk GTC DE genes in tumor cell clusters from (**b**). **d** Enrichment scores for bulk GTC DE genes and astrocyte-like tumor cell state markers in tumor cells from snRNA-seq data. **e** Log2-fold change of reactive astrogliosis markers between GTC-high and low samples in bulk RNA-seq data. Wald test, two-sided, fdr corrected. **f** Reactive astrogliosis marker expression in transcriptional tumor cell subpopulations from snRNA-seq data. **g** Differential protein presence in GTC-low (n = 53; negative log2FoldChange) and GTC-high (n = 7; positive log2FoldChange) ROIs from the NanoString GeoMx DSP proteomics data. Wald test, two-sided, fdr corrected. GTC: gemistocytic tumor cell; padj: adjusted p value; avg: average.

cell types (GTCs with T cells or TAMs with T cells) in both GTC high and low tumors (Fig. 5g). Next, we sought to assess how local CD44 densities compared to that of T cells or TAMs. To this end, we calculated normalized local CD44 kernel densities in whole slide images (see methods, Fig. 5h). To test which cell types increased alongside local CD44 density, all cell locations were assigned to a CD44 density value in each sample. Cell CD44 density values from all samples were pooled and separated into 20 bins (Fig. 5h). High CD44 density bins

were almost uniquely occupied by GTC-high samples and high T cell fractions exclusively increased in CD44 dense regions across GTC-high samples (Fig. 5i–k). TAMs were more widely present, yet their quantity showed a clear association with CD44 density in GTC-high samples. In contrast, T cells and TAMs showed no clear association with CD44 density in GTC-low samples. SPP1 and IL-1β are soluble markers and were not suitable for quantification with multiplex IF whole slide image analyses. We therefore used chromogenic

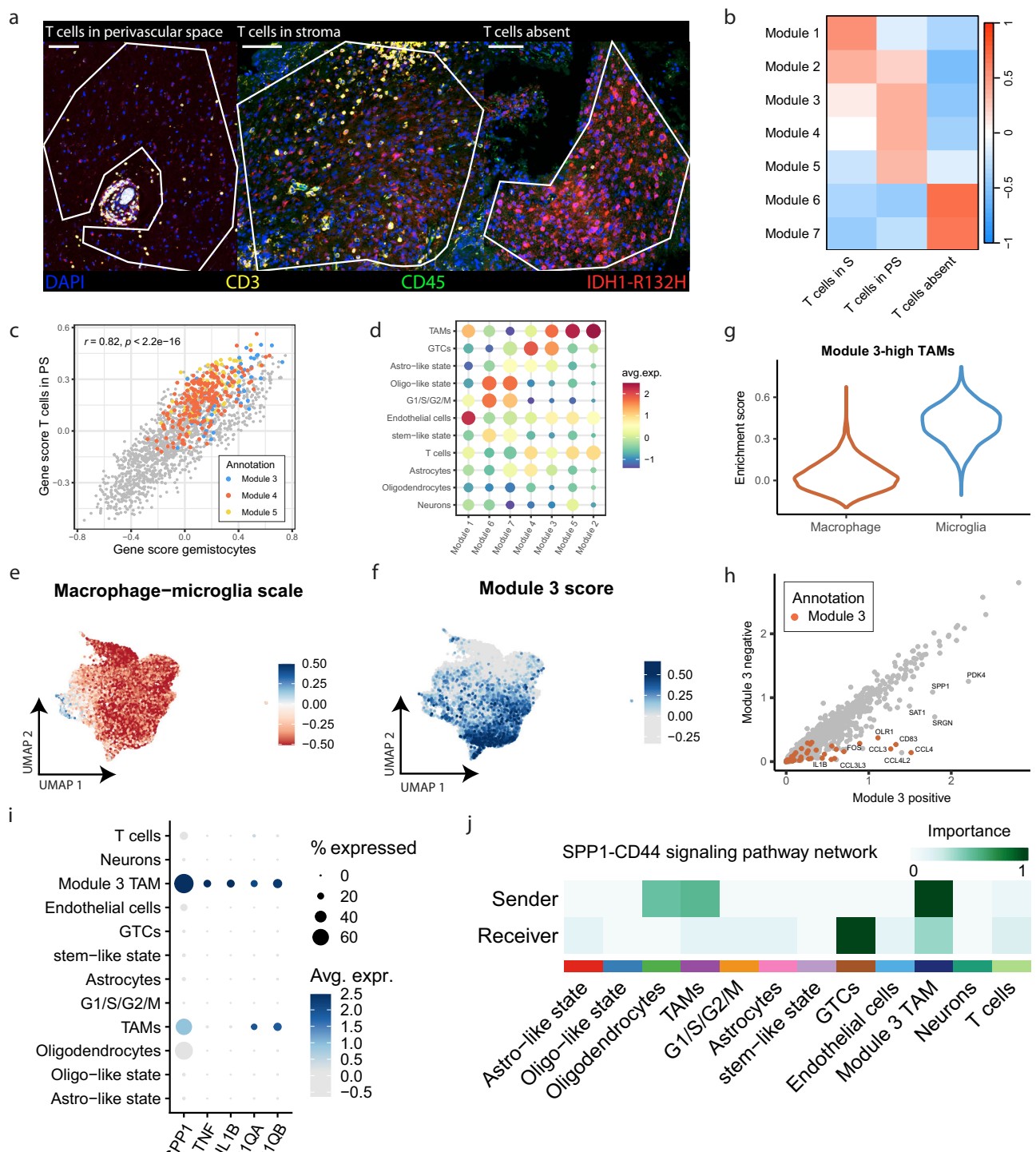

**Fig. 4 | Gemistocytic tumor cells co-localize with immune-reactive TAMs.**
**a** Examples of ROIs with annotations from the NanoString GeoMx spatial tran-
scriptomics data. For ROIs annotated as T cells in perivascular space, only stroma
was selected, circumventing the T cell cuff. **b** Heatmap showing correlation of
selected gene modules from (Supplementary Fig. 7d) with ROI annotation for
tissue phenotype. Pearson's r was used to determine correlation ($n = 25$ T cells in
PS; $n = 16$ T cells in S; $n = 27$ T cells absent). **c** Comparison between the gene sig-
nificance scores for T cells in perivascular space and GTC ROI annotations. Colors
indicate genes assigned to gene modules from b. Pearson's r, correlation t-test.
**d** Enrichment scores of spatial gene modules from b in snRNA-seq cell populations.
UMAP of TAMs depicting the difference in enrichment scores between monocyte

derived macrophage (MDM) and microglia markers (**e**) and enrichment scores for
gene module 3 (**f**). Blue indicates the MDM profile and red the microglia profile
($n = 10,775$). **g** Violin plot of enrichment scores for microglia and MDM markers in
module 3-high TAMs. **h** Comparison of average gene expression for Module 3
positive- and negative TAMs, split according to Module 3 enrichment score (Sup-
plementary Fig. 7g). **i** Expression of reactive astrogliosis signaling molecules in
snRNA-seq cell populations. **j** Cell-cell communication inference of SPP1-CD44
signaling in the snRNA-seq data. Green indicates the centrality score. Scale bars in a
indicate 100 μm. S: stroma; PS: perivascular space; TAM: tumor associated mac-
rophage; GTC: gemistocytic tumor cell; avg. expr: average expression.

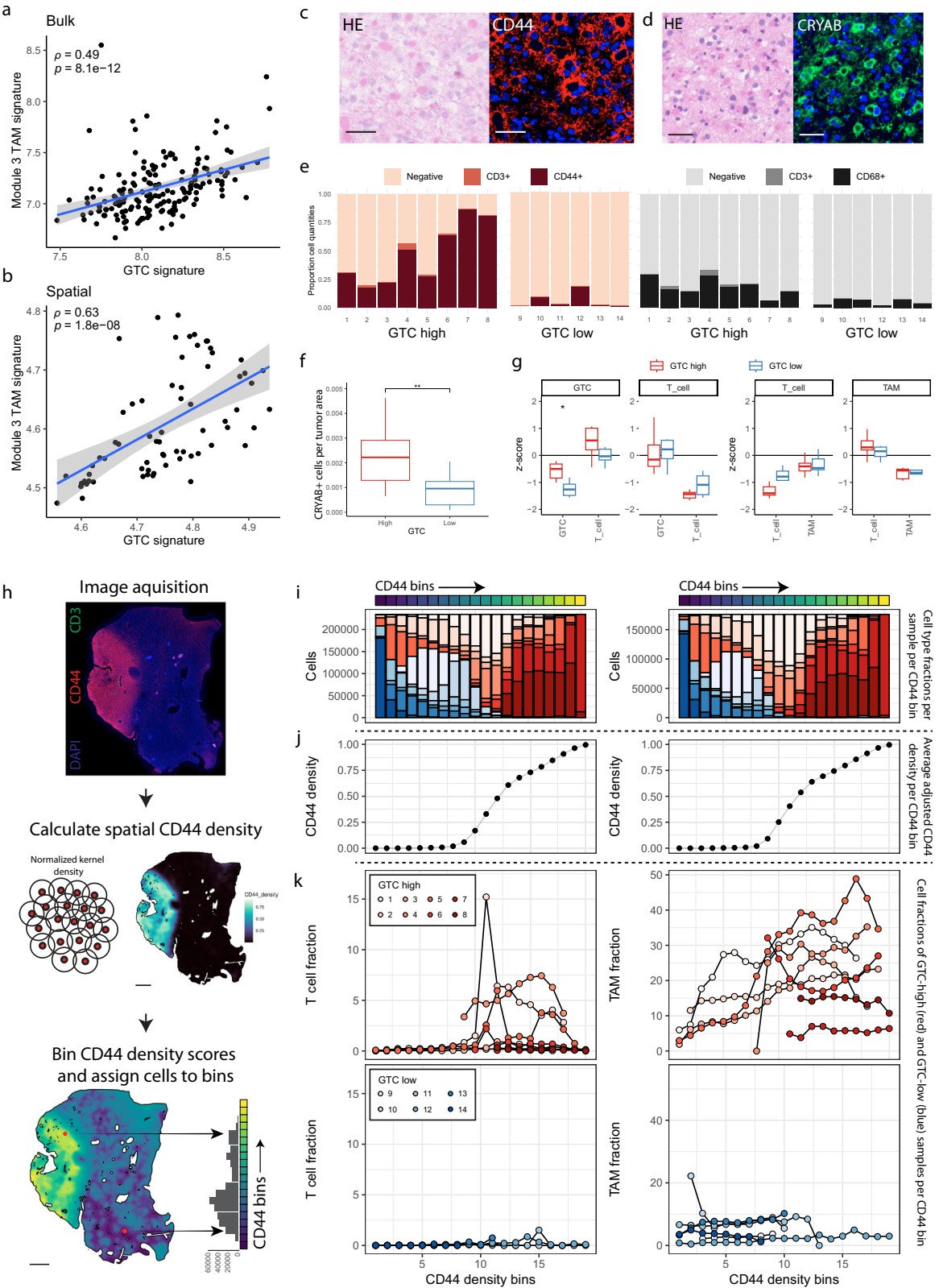

immunohistochemistry (IHC) stainings that showed presence of SPP1 and IL-1β in GTC-high regions and absence in GTC-low regions (Supplementary Fig. 8f, g). Collectively, our spatial and cell-cell interaction analyses in IDHmt astrocytoma show that GTCs and a subset of TAMs resembling immune-reactive microglia form a cellular network. Importantly, this network is a typical feature of the stromal surroundings of T cell cuffs and is phenotypically characterized by spatially restricted expression of candidate markers of glial scarring, such as CD44, SPP1, and IL-1β.

## Discussion

This study provides a large-scale and systematic evaluation of the IDHmt astrocytoma immune microenvironment and describes the cellular profiles underlying T cell exclusion. We demonstrate that even

**Fig. 5 | CD44-positive gemistocytic tumor cells form a network with immune-reactive TAMs that phenotypically characterizes the stroma surrounding T cell cuffs in IDHmt astrocytoma.** Correlation between average expression of snRNA-seq derived marker genes for GTCs and module-3 TAMs in bulk (**a**; $n = 169$) and spatial transcritptomics data (**b**; $n = 68$). Spearman's rho, correlation $t$-test. Examples of CD44 (**c**) and CRYAB (**d**) stainings with matching HE stainings of the same tissue slide. **e** Cell proportions for multiplex panels for GTC-high ($n = 8$) and GTC-low ($n = 6$) IDHmt astrocytoma samples. **f** CRYAB+ cell quantification in GTC-high ($n = 15$) and GTC-low ($n = 15$) IDHmt astrocytoma samples. Wilcoxon rank sum test, two-sided ($p = 0.00524$). **g** Inter-cell distance z-scores between cell types indicated in plot title (from) and cell type indicated in x-axis label (to) in GTC-high ($n = 8$) and GTC-low ($n = 6$) IDHmt astrocytoma samples. Wilcoxon rank sum test, two-sided, fdr corrected ($p = 0.0047$). **h** Workflow depicting CD44 density calculation and allocation of cells to CD44 bins in whole tumor sections. **i**–**k** Integrative analysis of two IF stainings (CD44/CD3 and CD68/CD3, left and right column, respectively) on consecutive sections of tumor samples ($n = 8$ GTC high, $n = 6$ GTC low). Detected cells from all samples were pooled and separated into 20 bins based on the adjusted CD44 density value at their tissue x/y coordinates. **i** Bar plot of the 20 equal-sized cell bins. The bars represent cell fractions from GTC-high (red) and GTC-low (blue) samples. **j** Average adjusted CD44 density per bin. **k** T cell (left) and TAM (right) proportion per bin for GTC-high (top) and GTC-low samples (bottom). GTC gemistocytic tumor cell. Scale bars in (**c**, **d**) indicate 50 μm. Scale bars in (**h**) indicate 2 mm. **a**, **b** show error bounds of ±SEM. Boxplots in (**f**, **g**) show the hinges at the first and third quartiles with the median as the center. The whiskers show min and max value until 1.5 times the interquartile range.

though IDHmt astrocytoma is largely considered deprived of T cells, these tumors contain T cell aggregates confined to the perivascular space around which GTCs co-localize alongside immune-reactive TAMs[5]. Our findings substantiate a model for T cell exclusion in IDHmt astrocytoma in which microglia produce SPP1 and IL-1β, thereby enabling crosstalk with CD44 and IL-1R-positive GTCs, a process that is highly reminiscent of glial scarring.

We were able to identify a GTC cell state using a bulk RNA-seq informed analysis of snRNA-seq data. Other single cell studies that evaluated IDHmt astrocytoma at single cell level did not discern a specific GTC cell state, which could be due to the targeted approach on the GTC cell state used in this study. Alternatively, a negative selection bias against large interconnected cells (like GTCs) in dissociation protocols used in most studies could limit their ability to detect GTCs[46]. GTC differentiation is recognized as a major tissue pattern in IDHmt astrocytoma according to the WHO classification for CNS malignancies (WHO 2021), but gets little attention due to a lack of clinical relevance[47]. The gemistocytic astrocytoma subclass has thus far been handled in a bulk fashion, disregarding tumor heterogeneity. Although some evidence already exists for a relationship between lymphocyte presence and GTCs in IDHmt astrocytoma, an in-depth examination of this immune phenotype was lacking[48,49].

Here, we made use of the GLASS-NL cohort that offers a unique opportunity to evaluate the TME across multiple resections and tumor grades[27]. Surprisingly, none of these factors affected the presence of the immune cells. Instead, we show that GTC presence is spatially associated with all immune cell subsets and that increased GTC density can be a focal occurrence. Moreover, GTC-high tumors were characterized by perivascular T cell cuffs, which is a known phenomenon in immune related CNS pathologies that form in the perivascular space between the endothelial and the parenchymal basement membrane[50]. In glioma, where the BBB is considered to lose its structural integrity, the formation of such T cell cuffs is more contentious[51]. However, the tumors we examined reliably showed an intact parenchymal basement membrane indicating that this barrier was not disrupted in vessels that host T cell cuffs. T cell localization does seem to differ between IDHmt and IDHwt tumors. In previous work, we showed that T cells were at a significantly farther distance from vessels in high-grade (WHO 2016 grade IV, predominantly IDHwt) compared to low grade tumors (WHO 2016 grade II/III). The reason for this difference remains to be determined but may be related to a difference in vascular structural integrity between IDHwt and IDHmt tumors[13]. It also remains unclear to what extent these lymphocyte clusters in the perivascular space can be considered organized structures as a more elaborate characterization of the leukocyte subsets is currently lacking. In other tumor types, such as melanoma, similar lymphocyte aggregates named tertiary lymphoid structures (TLS) mimic lymphoid organs and have been associated with an improved response to immune therapies[52]. In glioma, one study reported the formation of TLS structures in mice following agonistic CD40 therapy[53]. Although the lymphocyte aggregates described here cannot be formally classified as TLSs due to the lack of

typical CD4 + T cell and B cell organization, new treatments that would enhance T cell entry from these cuffs into the surrounding stroma could make tumors more prone to immune-mediated recognition and destruction.

We propose a model for confinement of T cells to the perivascular space, in which there is abundance of CD44 and IL1R1-expressing GTCs together with SPP1 and IL-1B-expressing TAMs in the surrounding stroma. These cell types constitute cellular networks and their gene activation programs resemble those found in reactive astrogliosis, a response of astrocytes to tissue damage which could lead to the formation of a glial scar[54,55]. During neuro-inflammation or degeneration in pathologies like MS, amyotrophic lateral sclerosis, stroke and Alzheimer's disease, microglia generally respond by secreting SPP1/ Osteopontin, IL-1β, and Complement (C)1q, which prompts quiescent astrocytes to acquire an immune-reactive phenotype[42,43,56,57]. Similar to the expression profiles in our study, these immune-reactive microglia express chemo-attractants such as CCL3 and CCL4[58]. CD44, which we identified as a marker for GTCs, can act as a receptor for SPP1[59]. An interaction between SPP1 and CD44 has previously been postulated for glioma, yet here we show an unexplored role of these molecules in conveying communication between GTCs, microglial cells and T cells[60–65]. The association between T cell cuffs and reactive astrogliosis further suggests that damage response processes are involved in repelling influx of T cells into the stroma. Notably, our TAM expression profiles were predominantly enriched for microglia markers, which is in line with reports showing that the majority of recruited TAMs are of microglial origin in IDHmt astrocytoma[5]. Although comparisons between reactive astrogliosis and other damage response processes should be done with some restraint, adoption of damage response-related processes is a more widely recognized mechanism of immune evasion and is reported in other tumor types as well[66–68]. Of note, GTC signature genes show overlap with the mesenchymal (MES) like cell state in glioblastoma and MES like cells have recently been linked to glial-like wound response pathways[69]. It is therefore possible that glial scarring mechanisms identified here are more generally present in primary brain tumors.

Collectively, our findings argue that T cell exclusion in IDHmt astrocytoma is caused by specific immunological determinants. In a mouse glioma model, induction of TLS formation and T cell accumulation but was unable to yield functional T cells nor enhanced responses to anti-PD1[53]. In IDHmt astrocytoma, T cell exclusion is typically accompanied by focal presence of GTCs, and even though the focal presence of GTCs is likely transient as evidenced by our analyses of consecutive resections, the early targeting of CD44+ GTCs or nearby SPP1+ TAMs holds potential therapeutic value to break immune tolerance in this disease. For example, antibodies targeting these cells could, depending on their drug-cargo, either destroy TAMs or skew these cells into a more pro-inflammatory phenotype. Such a sensitization strategy could render IDHmt astrocytoma amenable for anti-PD1 or other immune checkpoint inhibitors. Alternatively, identification of antigen targets specifically expressed by GTCs would enable the

development of adoptive T cell therapy to destruct these cells and overcome GTC-mediated T cell suppression. Currently, further research is required to intervene with glial scarring and develop therapeutics to enable T cell immunity in IDHmt astrocytoma.

Taken together, we have uncovered a cellular network between GTCs and reactive TAMs, potentially driven by reactive astrogliosis and glial scarring, that is a feature of IDHmt astrocytoma where T cells are confined to the perivascular space. Interference with this cellular network, resulting in the release of T cells, might offer a perspective for a better positioning of immune therapies to treat these tumors.

## Limitations to the study

With this study we aimed to characterize the immune system in the IDHmt astrocytoma TME. As such, we did not go into depth concerning functional T cell states that reside in the spatial phenotypes we describe here. Future studies that describe these states in relation to spatial phenotypes will be a vital next step in furthering our understanding of the relationship between IDHmt astrocytoma and the immune system as well as in identifying targets for new treatments.

Due to the nature of our data, we can only establish associations between the spatial phenotypes and transcriptional (tumor) cell states, and we can only make inferences about functional relationships. The mechanisms by which GTCs or T cell cuffs arise and how they can be manipulated need to be investigated functionally. Additionally, we have accumulated markers from a number of reports that list marker genes for reactive astrogliosis to be as complete as possible. Nevertheless, reactive astrogliosis is an incompletely understood and very diverse process that can have both pro- and anti-inflammatory effects[54]. Even though our results echo these pathways described in several pathologies, reactive astrogliosis and glial scarring need a more comprehensive assessment to solidify their exact phenotyping.

As we have reported previously, the NanoString GeoMx DSP data suffers from technical challenges that need extensive correction for its results to be reliable[33]. While we have shown that these data can be biologically meaningful, they should be interpreted carefully. To this end, we have contextualized and validated the results with other established techniques.

CD44 has been reported to be expressed by several cell types in the glioma TME, including immune cell subsets and glioma stem cells. It is therefore not an exclusive marker for GTCs and may cause noise in IF stainings. Regardless, CD44 was confidently identified as a marker for GTCs in all of our transcriptomic and proteomic data sets. Since SPP1 is a soluble factor, it proved difficult to quantify its presence in tissues. We therefore used single marker chromogenic IHC stainings for SPP1 instead of multiplex IF stainings to confirm the presence of SPP1 specifically in GTC-high tumors.

Nuances between the different data formats used in this manuscript can make comparisons between them difficult and should be performed with caution. First, ROI selection for spatial RNA and protein assays are not identical; ROIs annotated as T cell cuffs in the protein assay specifically select T cells, whereas the adjacent tumor stroma was selected in the RNA assay. Second, bulk and spatial expression profiles should be compared with some restraint, as the ROIs represent local biological processes within one tumor, and bulk measurements represent global biological patterns that might not represent these local expression patterns.

In this study, we used the GLASS-NL patient cohort that consists of tissue samples from consecutive resections of the same patients. This can complicate the interpretation of some results and introduce bias. Where possible, we corrected for this effect. Additionally, as this study only included IDHmt astrocytoma samples, results cannot be extrapolated to oligodendroglioma or glioblastoma. Lastly, the treatment of patients in this cohort was too heterogeneous to reliably assess its effect on the glioma TME.

## Methods

### Sample selection

All samples used in this study were IDH mutant astrocytoma. IDH-mutation status was determined for diagnostic purposes, using next-generation sequencing or IHC. Most samples were selected from the GLASS-NL study[27]. All patients provided informed written consent. These studies were approved by the ethical board of the Erasmus MC (MEC-2019-0288, Rotterdam, The Netherlands), and conducted in accordance with institutional and national regulations.

### Histological and immunohistochemical staining acquisition and assessment

**H&E staining and assessment.** H&E stainings were made according to standard practice for FFPE or cryopreserved tissue samples. FFPE and cryopreserved tissue sections were cut at 4 μm and 10 μm, respectively. H&E stainings were assessed by two independent reviewers for the abundance of GTCs under the supervision of a neuropathologist. Discrepancies were discussed with a third independent reviewer to reach a consensus. For the assessment of GTC abundance in ROIs from multiplex IF stainings, the same tissue slide was used for H&E stainings where possible. Consecutive slides were used for H&E staining of all other ROIs. 334 ROIs were excluded because the H&E staining could not be assessed or regions were missing on consecutive slides. For NanoString GeoMx DSP (NanoString, Seattle, WA, USA) spatial transcriptomics and proteomics ROIs, consecutive slides were used. Regions on HE stainings were annotated to match the IF ROIs. All ROIs were separated into four bins depending on relative GTC quantity: 0%; 0–20%; 20–40% and >40%. For bulk RNA sequencing samples, hole-slide H&E stainings were performed on tissue sections consecutive to those used for sequencing and were assessed for GTC abundance. These samples were categorized as GTC-high or low using a threshold of ~20% GTC of all malignant cells according to WHO 2021. For two-group DE testing and comparative analysis with bulk RNA sequencing, NanoString GeoMx DSP GTC ROI annotations were separated into two bins with a threshold of 20%. All scoring was done blinded to the IF stainings and clinical annotations of the samples.

**Multiplex immunofluorescence stainings.** Multiplex IF stainings were performed on using either Opal™-tyramide signal amplification multiplex immunohistochemistry (Akoya Biosciences, Marlborough, MA, USA) or conventional fluorophore tagged secondary antibodies. Stainings using the Opal™ reagents were performed as follows: Antigen retrieval using AR9 (Akoya Biosciences, #AR900250ML) at 90 °C for 15 min was followed by incubation in antibody diluent (ABD; Akoya Biosciences, #ARD1001EA) for 45 min at room temperature (RT), anti-CD56 (1/500 in ABD; Cell Marque, Rocklin, CA, USA, #MRQ-42) staining for 30 min at RT, incubation with HRP polymer Ms + RB (1/500 in ABD; Akoya Biosciences, #ARH1001EA) for 10 min at RT and finally incubation with opal 620 (1/100 in ABD; Akoya Biosciences, #OP-001004) for 10 min at RT. Antigen retrieval using AR9 at 90 °C for 15 min was followed by incubation in ABD for 10 min at RT, anti-CD3 (1/100 in ABD; Sigma-Aldrich, Saint Louis, MO, USA, #SAB5500058) staining for 30 min at RT, incubation with HRP polymer Ms + RB (1/500 in ABD) for 10 min at RT and finally incubation with opal 520 (1/100 in ABD, Akoya Biosciences, OP-001001) for 15 min at RT. Antigen retrieval using AR6 (Akoya Biosciences, #AR600250ML) at 90 °C for 15 min was followed by incubation in ABD for 10 min at RT, anti-CD20 (1/1000 in ABD; Perklin Elmer, Waltham, MA, USA, #OP4LY2001KT) staining for 30 min at RT, incubation with HRP polymer Ms + RB (1/500 in ABD) for 10 min at RT and finally incubation with opal 650 (1/100 in ABD; Akoya Biosciences, #OP-001005) for 10 min at RT. Antigen retrieval using AR9 at 90 °C for 15 min was followed by incubation in ABD for 10 min at RT, anti-CD8 (1/500 in ABD; Cell Marque, #108M-96) staining for 30 min at RT, incubation with HRP polymer Ms + RB (1/500 in ABD) for 10 min at RT and finally incubation with opal 570 (1/100 in ABD; Akoya

Biosciences, #OP-001003) for 10 min at RT. Antigen retrieval using AR6 at 90 °C for 15 min was followed by incubation in ABD for 20 min at RT, anti-CD68 (1/500 in ABD; Cell marque, #168M-96) staining for 30 min at RT, incubation with HRP polymer Ms + RB (1/500 in ABD) for 10 min at RT and finally incubated with opal 540 (1/200 in ABD; Akoya Biosciences, #OP-001002) for 10 min at RT. Antigen retrieval using AR6 at 90 °C for 15 min was followed by incubation in ABD for 10 min at RT, anti-IDH1-R132H (1/20 in ABD; Dianova, Geneva, Switserland, #DIA-H09) staining for 45 min at RT, incubation with HRP polymer Ms + RB (1/500 in ABD) for 20 min at RT and finally incubation with opal 690 (1/100 in ABD; Akoya Biosciences, #OP-001009) for 10 min at RT. Finally, nuclei were stained with DAPI (Akoya Biosciences, #FP1490). PBST was used for all wash steps.

Conventional multiplex IF stainings using fluorophore tagged secondary antibodies were performed using the Ventana Benchmark Discovery ULTRA (Ventana Medical Systems, Oro Valley, AZ, USA). An overview of the samples used for multiplex IF stainings can be found in Supplementary Data 7. Consecutive 4 µm FFPE sections were stained for 4 panels: CD31, CD3 and LAMA2; CD3, CD68, IDH1-R132H and CD31; CD3, CD44, SPP1 and TNC; CRYAB. Slides were imaged with the LSM 700 (Zeiss, Oberkochen, Germany) or the Axioscan (Zeiss). For all panels, deparaffinization and heat-induced antigen retrieval with CC1 (Ventana, #950-224) for 32 min was performed at 100 °C. For the first panel, samples were first incubated with anti-CD3 (2.05 µg/ml, Ventana, #2GV6) for 32 min at 37 °C followed by Omnimap anti-rabbit HRP (Ventana, #760-4311) and detection with Red610 (Ventana, #760-235) for 8 min. Antibody denaturation was performed with CC2 (Ventana, #950-123) at 100 °C for 20 min. Second, incubation with anti-LAMA2 (1/100; Sigma Aldrich, #5H2) was performed for 32 min at 37 °C, followed by Omnimap anti-mouse (Ventana, #760-3210) and detection with Cy5 (Ventana, #760-238) for 8 min. Last, incubation with anti-CD31 (1.04 µg/ml; Cell marque, #JC70) was performed for 32 min at 37 °C, followed by Omnimap anti-mouse and detection with FAM (Ventana, #760-234) for 8 min. For the second panel, samples were first incubated with anti-CD3 (2.05 µg/ml) for 32 min at 37 °C followed by Omnimap anti-rabbit HRP and detection with Cy5 for 8 min. Antibody denaturation was performed with CC2 at 100 °C for 20 min. Second, incubation with anti-CD68 (3.49 µg/ml; Ventana, #KP1) was performed for 20 min at 37 °C, followed by Omnimap anti-mouse HRP and detection with DCC (Ventana, #760-244) for 8 min. Antibody denaturation was performed with CC2 at 100 °C for 20 min. Third, incubation with anti-IDH1-R132H (1/800) followed for 32 min at 37 °C, followed by Omnimap anti-mouse and detection with FAM for 8 min. Antibody denaturation was performed with CC2 at 100 °C for 20 min. Last, samples were incubated with anti-CD31 for 32 minutes at 37 °C, followed by Omnimap anti-mouse and detection with Red610 for 8 min. For the third panel, samples were first incubated with anti-CD3 (2.05 µg/ml) for 32 min at 37 °C followed by Omnimap anti-rabbit HRP and detection with FAM for 8 min. Antibody denaturation was performed with CC2 at 100 °C for 20 min. Second, incubation with anti-CD44 (1/8000; Novusbio, Centennial, CO, USA, #8E2F3) was performed for 32 min at 37 °C, followed by Omnimap anti-mouse and detection with Red610 for 8 min. Antibody denaturation was performed with CC2 at 100 °C for 20 min. Third, samples were incubated with anti-SPP1 (1/800; Santa Cruz, Dallas, TX, USA, #D2723) for 32 min at 37 °C, followed by Omnimap anti-mouse and detection with DCC for 8 min. Antibody denaturation was performed with CC2 at 100 °C for 20 min. Last, samples were incubated with anti-TNC (1/500) for 4 min at 37 °C, followed by Omnimap anti-mouse and detection with Cy5 for 8 min. For the fourth panel, samples were incubated with anti-CRYAB (1/1000, Abcam, #1B6-13g4) for 32 min at 37 °C followed by Omnimap anti-rabbit HRP and detection with FAM for 8 min. Finally, slides were washed in phosphate-buffered saline and mounted with Vectashield (Vector Laboratories, Newark, CA, USA, #H-1200-10) containing 4′,6-diamidino-2-phenylindole.

**Immunohistochemistry for VIM, IL1β and SPP1.** Two GTC-high and two GTC-low samples were stained for VIM (#347M-10), IL-1β (Abcam, #ab9722) and SPP1 (Santa Cruz, #D2723) by automated IHC using the Benchmark ULTRA (Ventana). Sequential 4 µm thick (FFPE) sections were stained for VIM, IL-1β or SPP1 using the Optiview DAB IHC Detection Kit (Ventana, #760-700). Following deparaffinization and heat-induced antigen retrieval with CC1 for 32 min, the tissue samples were incubated with VIM, IL-1β or SPP1 primary antibodies (1.04 µg/ml, 1/3200 and 1/800, respectively) for 32 min at 37 ˚C. Incubation was followed by Optiview detection and hematoxylin II counter stain for 8 min followed by a blue coloring reagent for 8 min according to the manufacturer's instructions (Ventana).

**Electron microscopy.** Cryopreserved GTC-high samples were selected based on H&E stainings. Samples were fixed in a solution of 25% glutaraldehyde and 37% formaldehyde and dehydrated in a series of acetone solutions using the EM TP Automated Tissue Processor (16709202, Leica, Wetzlar, Germany). Samples were impregnated with an epoxy solution (manufacturer procedures were followed; Embed 812 (Aurion, Jalandhar, India, #14901); NMA (Aurion, #19001); DMP-30 (Aurion, #13600); DDSA (Aurion, #13711)) for one hour at 45 °C. Tissue sections of 1 µm were cut and stained with Toluidine blue (0672-25 G, VWR) to confirm presence of GTCs. Ultrathin sections of 50 nm to 70 nm were cut and mounted on a Transmission Electron Microscopy copper grid (Aurion, # G200-Cu). Samples were contrasted on the EM AC20 (Leica) with Uranyless (Aurion, #22409-20) and Ultrostain II (Aurion, #NC1936447). Samples were imaged using the JEM-1011 Transmission Electron Microscope (JEOL Ltd., Akishima, Japan).

### Multiplex immunofluorescence staining image analysis

**Opal based IF Image analysis.** Following whole slide scans using VECTRA 3.0 (Akoya Biosciences), at least eight ROIs (size: 670 × 502 µm²; pixel size: 0.5 × 0.5 µm²) were equally divided over tumor cell-high and tumor cell-low regions as well as over immune cell-high and immune cell-low regions. The selection of ROIs in a given sample was performed unbiassed to any clinical patient or molecular parameter. In case parts of the tissue were disrupted or lost due to repeated staining cycles, fewer stamps were set or tissues were excluded from analysis (if stamp $n < 3$). Images were spectrally unmixed using inForm® software (Akoya Biosciences) to visualize markers of interest as well as auto fluorescence. Images were analyzed using the Python based TME-Analyzer software[70]. Each image was processed individually with the following steps; (i) tissue foreground was determined as regions positive for signal in any channel, with the exception of the autofluorescence channel, through manual thresholding; (ii) nuclei detection was performed in the foreground area using the Stardist algorithm and (iii) cell detection through expansion of detected nuclei region by 30 pixels (15 µm); (iv) cells phenotyping was performed manually according to their fluorescent intensities for each marker as measured in whole cells, nuclei or cytoplasm. Following image analysis, cell numbers and distances were calculated. The distance from a cell with a certain phenotype to the nearest cell with another phenotype was calculated by NN analysis and averaged for all cells in individual ROIs. Z-scores were consequently determined by randomly reassigning labels 1000 times to acquire a normal distribution of random NN distances for the cell type numbers within the image and comparing the NN value with the normal distribution[70]. As all ROIs had the same dimensions, normalization to tissue area would not affect results. Values from individual ROIs and mean values per resection were used for visualization and statistical testing. The cutoff for tumor high- and low regions was set at the median for tumor cell count in all ROIs. For statistical testing and visualization, a pseudo-count was added and counts per ROI were log-transformed.

**Conventional multiplex IF staining image analysis.** Whole slide image analysis of validation stainings were performed using QuPath[71] and R[72]. For combined analysis of the panels CD44/CD3 and CD68/IDH1-R132H, one sample was excluded because it did not have an IDH mutation and five samples did not pass QC due to bad signal quality (Supplementary Data 7). For quantification of CD31/CD3/LAMA2, two sample did not pass QC. First, whole slide scans were processed in QuPath, where tissue regions were defined using average signal intensity from all channels, and artifacts were manually excluded. Within tissue regions, cell detection was performed using the StarDist extension in QuPath. The standard dsb2018_heavy_augment model was used. NormalizePercentiles were set to 0.1 and 99.9, tileSize was set to 1500, Probability threshold was set to 0.5, cellExpansion was set to 5 μm, and cellContrainScale was set to 1.5. Methods were identical for all whole slide image analyses. For CD3/CD31/LAMA2 multiplex IF stainings, CD31 and LAMA2 positive objects were detected by training a neural network for pixel classification in QuPath separately for every sample.

Results were consequently exported as GeoJSON objects and imported into R. Signal thresholds were defined for all markers individually for every sample, blinded for sample type (Supplementary Data 7). The mean cytoplasm signal was used for CD44, mean nucleus signal for CD3, and max cytoplasm signal for CD68. Cells that were double positive for CD3 and CD44 were annotated as T cells. Tissue region polygons were used as the window and the mean XY locations of the annotated cell polygons were used to construct annotated planar point pattern objects that were analyzed using Spatstat, Jsonify and sf R packages[73]. Z scores for NN analysis were calculated using the nndist function from the Spatstat package. First, for every sample, cell labels were randomly reassigned 100 times and NNs were calculated. Next, actual NN distances were calculated. Z score was calculated by comparing the mean of NN results to the mean and the standard deviation of the random NN analysis. To compare consecutive slides in R, a consensus tissue outline was acquired. The overlay of two tissue regions was first maximized by rotation and adjustment of their XY coordinates. The consensus tissue outline was constructed by including only regions that were overlapping between the two tissue outlines. Using this outline, the adjusted CD44 kernel density was compared with CD68+ cells.

Kernel densities for CD44 positive cells and all cells were calculated for whole slide scans using the *density* function in the Spatstat R package with the sigma parameter set to 500. The ratio of CD44+ cell kernel density over total cell kernel density was used as the corrected CD44 density measure. The corrected CD44 density was determined for all cells based on the joint XY coordinates. Based on adjusted CD44 density, the cells from all samples were separated into 20 bins that all contained an equal number of cells. The fraction of T cells and TAMs was calculated per bin and per sample.

LAMA2 positive objects were converted into nonconvex hulls using the fm_nonconvex_hull function from the *fmesher* R package with convex set to −0.001 and concave to 100. Cells were assigned to a LAMA2 convex hull if its centroid was contained by it. T cell cuffs were defined as convex hulls containing more than 2 T cells. Distance from T cells to the nearest CD31+ blood vessel object was calculated using the st_nearest_feature and the st_distance function from the sf R package. Within all samples, a maximum of 2000 T cells and negative cells were sampled for quantification.

**Methylome, transcriptome and proteome data acquisition**
**Bulk RNA sequencing.** The bulk RNA sequencing data from the GLASS-NL cohort was used (sample details can be found in Supplementary Data 2)[27]. Briefly, FFPE samples were selected for high tumor cell content (>50% as assessed by a pathologist). Macrodissection was performed on 10 μm sections and RNA was isolated using the RNAeasy FFPE isolation kit (Qiagen, Hilden, Germany, #73504; according to

manufacturer protocols). After cDNA synthesis and amplification, paired-end sequencing was done on the NovaSeq6000 (Illumina, San Diego, CA, USA). After QC, reads were aligned to GRCh38, filtered for duplicate reads and feature counts were extracted.

**Bulk methylation profiling and CNV inference.** We used the bulk methylation profiling data from the GLASS-NL cohort[27]. Briefly, 80–250 μg DNA was bisulfite converted using the EZ DNAmethylation Kit (Zymo Research, Irvine, CA, USA) and processed on Infinium MethylationEPIC BeadChip arrays (Illumina). We inferred the copy number profile of GLASS-NL samples used in our snRNA-seq analysis from the DNA methylation data. The CNV profiles were estimated by using the igv files (cnvp v5.2) generated after uploading the raw idat files in the Heidelberg CNS tumor classifier (v12.8)[74]. In brief, log2 intensity differences for each genomic segment were calculated by comparing the mean methylation probe intensity value against healthy reference samples with a flat CNV profile. The resulting copy-number profile was then plotted according to its chromosomal location.

**Single nucleus RNA sequencing.** Seven tumor samples from six patients of the GLASS-NL cohort were selected for snRNA sequencing (sample details can be found in Supplementary Data 4). Nuclei isolation and sequencing were performed as described previously[75]. Briefly, Cryopreserved samples were taken from −80 °C storage. For all samples, an H&E staining was made to assess the tumor content and the quality of the tissue. OCT (Sakura Finetek, Torrance, CA, USA, #4583) was removed as much as possible using a scalpel and a sample of -0.1 g was taken for processing. Tissue was washed with ice-cold PBS to remove residual OCT. Next, the tissue was suspended in 1.5 ml lysis buffer (EZ lysis buffer (Merck, Rahway, NJ, USA, #NUC101); 2 U/ml RNAse inhibitor (RNaseOUT, Thermo-Fisher, Waltham, MA, USA, #10777019)) in an all glass tissue grinder (DWK Life Sciences, Wertheim, Germany, #885300-0002). The two pestles were plunged 25 times each to release nuclei and the suspension was incubated on ice for 5 min. Next, the suspension was moved to a 15 ml tube, 1.5 ml lysis buffer was added and nuclei were incubated for another 5 min. Nuclei were filtered through a 70 μm cell strainer and centrifuged at 500 g for 5 min in 4 °C. Supernatant was discarded and nuclei were resuspended in 2–3 ml of lysis buffer depending on the pellet size and incubated on ice for 5 min. Next, nuclei were centrifuged at 500 g for 5 min in 4 °C, supernatant was discarded and 1 ml of wash and resuspension buffer (WRB) (1x PBS 10% BSA, 2 U/ml RNAse inhibitor, Hoechst) was added without disturbing the pellet and was incubated for 5 min to let the buffer interchange. Another 2 ml of WRB was added, nuclei were resuspended and filtered through 40 μm cell strainer. Fluorescence activated nuclei sorting (FANS) was used for QC and for debris removal. Next, nuclei quality and quantity was assessed using a counting chamber and after resuspension of the correct concentration per μl, nuclei were processed on the 10x Genomics single-cell RNA sequencing platform using 10x Chromium Single-cell library preparation protocols.

**NanoString GeoMx DSP RNA and protein profiling.** FFPE samples from 12 paired tumor resections of six IDHmt astrocytoma patients from the GLASS-NL cohort were selected for spatial RNA and protein profiling (Supplementary Data 3, 5). Regional expression data were collected using the GeoMx DSP system (NanoString, Seattle, WA, USA). Sample processing and probe collection were performed by NanoString. Tumor material from both resections were placed on the same slide for every patient to avoid intra-patient batch effects. For both RNA and protein, 12 regions per patient and -6 regions for every resected tumor were selected based on IF staining for CD3, CD45, IDH1-R132H and DAPI. Regions were selected for the T cell phenotypes that were present; T cells in perivascular space, T cells in stroma and T cell absent (Supplementary Data 3, 5). For T cell cuffs, ROI selection

differed between the RNA and protein assays. For RNA, only the stroma around T cell cuffs were selected and T cell aggregates were avoided. For protein, only T cell cuffs were selected, with minimal stroma included. For RNA, 8684 probes coupled to a photocleavable oligonucleotide tag were used to measure the expression of 1825 genes (3–5 probes per target). For protein, antibodies coupled to a photocleavable oligonucleotide tag were used to measure the presence of 54 proteins.

### Transcriptomics and proteomics data preprocessing

**Bulk RNA-sequencing.** Lowly expressed genes with an average read count of less than three per sample were excluded. Only genes with GENCODE V34 transcript annotation protein-coding and long non-coding RNA were considered. Raw bulk RNA-seq data were normalized and VST transformed according to the DESeq2 methods.

**Single nucleus RNA-sequencing.** SnRNA-seq data was preprocessed using CellRanger mkfastq and Cellranger count V3. Unspliced reads were used for alignment to GRCh38. Quality measures for all samples can be found in Supplementary Data 3. Samples were further processed in R using the Seurat package[76]. Doublets were identified and excluded using the function *computeDoubletDensity* from the scDblFinder R package[77]. Doublet scores were log-transformed and a cutoff was determined by visual inspection of the score density for all GEMs and the distribution of the doublet score in a Seurat feature plot. All samples were normalized using the *SCTransform* V2 function. Data was integrated using reciprocal PCA as integrated in the Seurat package. For analysis on tumor cells and TAMs, the cell populations were identified in the integrated dataset, filtered from individual samples and reintegrated separately.

**NanoString GeoMx spatial RNA and protein.** NanoString GeoMx spatial transcriptomics and proteomics data QC was performed as described previously[33]. Probe count collection, quality control and reference mapping of reads was performed according to manufacturer protocols (MAN-10119-01 GeoMx-NGS Data Analysis User Manual). 1673 genes and 65 ROIs passed QC for the RNA panel. 41 proteins and 69 ROIs passed QC for the protein panel. Spatial transcriptomics data was normalized using quantile normalization[33]. Spatial proteomics data was normalized using negative control Spike-in probes according to manufacturer protocols.

### Transcriptomics and proteomics data analysis

**Bulk RNA-sequencing.** DGE testing was performed using the DeSeq2 R package[78]. We applied batch-correction to account for multiple resections from the same patient by including this variable as a covariate in the design formula in DESeq2 in the bulk RNA-seq DE test. Markers for TAMs and T cells were taken from previously published work (Supplementary Data 10)[61,79,80]. Bulk DE genes for GTC-high and GTC-low samples were used for pathway enrichment analysis in Ingenuity Pathway Analysis (IPA) software (Qiagen). The log2fc cutoffs were set at -0.5 and 0.5 and the adjusted *p*-value cutoff at 0.001. Otherwise, standard settings were used.

**NanoString GeoMx DSP spatial transcriptomics and proteomics.** Differential expression tests for Nanostring GeoMx DSP spatial transcriptomics and proteomics were performed using the DESeq2 R package[78]. Normalized counts of the NanoString data were rounded and sizeFactors were set to 1 to perform DGE tests based on ROI GTC scores. Separately, unbiased clustering of NanoString GeoMx DSP protein data was performed using the ComplexHeatmap R package. Weighted Gene Co-expression Network Analysis (WGCNA) was performed using the WGCNA R package to construct gene modules using the expression data[33]. As one sample was a consistent outlier in gene correlation analyses and had an outlier profile in WGCNA

preprocessing, we excluded it from all analyses. Quantile normalized and log2-transformed data were used as input. First, a similarity matrix was constructed of the data using the Pearson correlation. Next, the similarity matrix was transformed into a signed weighted adjacency matrix using a soft thresholding power of 16 that was determined by approximation of the scale-free topology criterion. From this, the interconnectedness of all genes was assessed using the topological overlap measure. Using hierarchical clustering for the topological overlap matrix, modules were defined using the Dynamic Tree cut algorithm (as integrated in the WGCNA package) with a DeepSplit parameter of 3 and a minimum module size of 30. Gene importance scores were defined as the Pearson correlation of module eigengenes with the normalized gene expression profiles. Gene modules were correlated with sample metadata using Pearson's r as standard in WGCNA processing.

**Single nucleus RNA-sequencing.** Single nuclei from snRNA-seq results were assigned to cell types using marker genes for non-malignant cell types and malignant transcriptomic cell states[34,79,80]. Nuclei that could not reliably be assigned to a cell cluster were excluded from further analysis. The enrichment score of a nucleus for a marker gene set was assessed as follows: for a gene set G a reference gene set R was constructed. This was done by binning the all genes from the expression dataset into 30 equal-sized bins according to expression level. For every gene in G, 100 random genes from the corresponding bin were added to R, resulting in two gene sets, so that R was 100x larger than G. The difference in average expression for G and R was consequently calculated for each cell to provide an enrichment score. Thresholds for enrichment scores were assessed based on visual inspection of its distribution. DE genes for cell clusters in snRNA-seq data were determined using the *FindMarkers* function from the Seurat R package. The complete snRNA-seq dataset was used to find DE genes. Copy number variations were inferred on snRNA-seq data using the InferCNV package[81]. Standard parameters for 10x genomics samples were used. For every tumor, cell types were identified as described above. Consequently, non-malignant cell clusters were used as a reference for cell clusters that showed tumor cell expression profile. Pre-noise filtering estimate objects from the InferCNV output were used for visualization. For each tumor, cells were grouped according to estimated transcriptional tumor state. Markers for reactive astrocytes were collected from literature without any further filtering or selection (Supplementary Data 11)[40–43]. For enrichment calculations, overlapping genes between marker gene sets and the top 2000 most variable genes from the respective snRNA-seq datasets were used. Cell-cell interaction inference was performed in our integrated snRNA-seq dataset using the CellChat.DB R package. We included all standard cell-cell interaction pathways that are included in the CellChat.DB package. Standard inference analysis was used with the default 25% truncated mean (triMean) setting[82]. Centrality scores for annotated cell types are a measure of importance in a specific pathway as a "sender" or "receiver" and was adopted from the standard CellChat.DB output.

### Statistical analysis

Most statistical analyses were performed in R. Wald's test was used for DGE testing for bulk RNA and spatial RNA and protein analysis as implemented in DeSeq2. DE genes for cell clusters in snRNA-seq data were determined using the FindMarkers test from the Seurat R package which uses a likelihood ratio test. A right-tailed Fisher's Exact Test was used for statistical testing of the pathway enrichment analysis in IPA. All other statistical tests used are specified in the results section. Statistical significance was defined as a *p*-value of < 0.05 for single comparison tests or adjusted *p*-value of < 0.05 for multiple comparisons. *P*-values were adjusted using fdr correction.

**Reporting summary**

Further information on research design is available in the Nature Portfolio Reporting Summary linked to this article.

## Data availability

VECTRA ROI images are available on Zenodo and processed count and cell-cell distance values can be found in supplementary documents (https://zenodo.org/records/13911692, 10.5281/zenodo.13911692; Supplementary Data 9). Processed count data from 10x single nucleus RNA sequencing experiments are available on Zenodo (https://zenodo.org/records/10435521, https://doi.org/10.5281/zenodo.10435521). NanoString GeoMx spatial transcriptomics data is available on Zenodo and sample annotations are provided in supplementary documents (https://zenodo.org/records/13911761, https://doi.org/10.5281/zenodo.13911761; Supplementary Data 1)[33]. NanoString GeoMx spatial proteomics data can be found in the supplementary documents, ROI images are available at Zenodo (https://zenodo.org/records/13918032; Supplementary Data 6; https://doi.org/10.5281/zenodo.13918032). Whole-slide scans of multipex IF stainings and corresponding GeoJSON and measurement files are available on Zenodo (CRYAB/HE: https://zenodo.org/records/13918123, https://doi.org/10.5281/zenodo.13918123; CD31, CD3, LAMA2: https://zenodo.org/records/13918377; https://doi.org/10.5281/zenodo.13918377; CD44, CD3, SPP1, TNC and CD68, IDH1-R132H, CD3, CD31: https://zenodo.org/records/13911899, https://doi.org/10.5281/zenodo.13911899; Supplementary Data 7). Bulk RNA sequencing data are available at the European Genome-Phenome Archive (https://ega-archive.org/studies/EGAS00001007551, EGAD50000000558).

## Code availability

Code used for processing, analysis and visualization can be found on GitHub (https://github.com/LevivanHijfte/GTC_paper, https://doi.org/10.5281/zenodo.14500840).

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

## Acknowledgements

This research was funded by Het Hersentumorfonds (DBTF-RG201901).

## Author contributions

Conceptualization, L.H., M.G., R.D. and P.F.; Methodology, L.H., M.G., S.A.G., H.E.B. and Y.H.; Software, L.H. and H.E.B.; Formal Analysis, L.H., I.H., S.A.G., R.L., H.E.B. and Y.H.; Investigation, L.H., M.G., I.H. and R.H.; Resources, S.A.G., W.R.V., B.W., P.W., T.B. and H.E.B.; Data Curation, L.H., M.G., I.H., W.R.V. and Y.H.; writing – original draft, L.H., M.G., R.D. and P.F.; writing – review & editing, L.H., M.G., I.H., S.G., H.E.B., Y.H., W.R.V., R.H., T.B., B.W., P.W., J.A.J., R.D. and P.F.; visualization, L.H., I.H., S.A.G. and Y.H. supervision, M.G., R.D. and P.F.; funding acquisition, R.D. and P.F.

## Competing interests

The authors declare no competing interests.
