## [Transparent Peer Review file · Nature Communications]

Gemistocytic tumor cells programmed for glial scarring characterize T cell confinement in IDH-mutant astrocytoma

Corresponding Author: Mr Levi van Hijfte

Version 0:

Reviewer comments:

Reviewer #1

(Remarks to the Author)

The authors have studied IDH mutant astrocytoma, which has been long considered a T cell-deprived tumor. They found that, when present, T cells are either distributed in the tumor stroma, or in the majority of cases, contained in the perivascular space (i.e., T cell cuffs). Specifically, they found that gemistocytic tumor cells (GTCs) constitute a distinct sub-cluster of the astrocyte-like tumor cell state that co-localize with microglia expressing the chemo-attractants CCL3 and CCL4. Neighboring GTCs and microglia express receptors and ligands that typically take part in reactive astrogliosis and glial scarring, such as Osteopontin (SPP1) – CD44 and IL-1 β – IL1R1. The authors have identified a cellular network of GTCs and microglia that show features of glial scarring, a central nervous system specific mechanism of T cell exclusion, that features in IDHmt astrocytomas where T cells are confined the perivascular space.

The work is well written. I have a few specific comments below:

1. Figure 1 needs more detailed and to the point analysis to demonstrate that T-cells cluster around blood vessels. They should measure distance from blood vessels in T-cells compared to other cell types in the tumor and should compare this in IDH-mut vs. IDH-wt tumors. Similarly, they should look at what % of tumor samples are considered "T-cells in perivascular space" vs. "T-cells in stroma" or "T-cells absent" in IDH-mut vs. IDH-wt gliomas.
2. Did the % of tumor samples considered "T-cells in perivascular space" change with WHO grade or with recurrence vs newly diagnosed?
3. The authors need to introduce gemistocytic astrocytomas in more of an unbiased fashion with some background for those not familiar with this concept. They should perform EM analysis of the tumor samples and describe different morphologic features they see, including gemistocytic and whatever term(s) they would like to use for the non-gemistocytic tumor cells (terms like fibrillary and protoplasmic have been used in the past but are no longer recognized so it would be reasonable to just classify as gemistocytic vs. non-gemistocytic).
4. The authors should use the cell chat algorithm on their single nuclear results to look at tumor cell communication with macrophages and T-cells in the perivascular space vs. the stroma to investigate their hypothesized mediators.
5. Figure 4a spatial transcriptomics please formally define the region of what is the perivascular space and what is the T-cell cuff. We need to know exactly how close the stroma was for ROIs annotated as T cells in the perivascular space to be certain that these samples are distinct from "T-cells in the stroma" samples.
6. The authors used CD44 as a GTC marker but it is also an endothelial marker and endothelial cells are close spatially to GTCs. Endothelial cells can also have CNV changes in GBM so an additional marker will be needed to be confident that these are gemistocytic tumor cells.
7. Were any canonical GBM mutations (e.g. TERT promoter, CKDN2A/B, 1p/19q, EGFR) more common in gemistocytic vs. non-gemistocytic tumor cells?

Reviewer #2

(Remarks to the Author)

This manuscript focuses on using bulk, single nucleus, spatial transcriptomics sequencing and proteomics to study IDH1 mutant astrocytoma. They found an immune phenotype that is based on T-cell exclusion in T-cell cuffs characterized by the presence of GTC cells.

I highly recommend the authors add a “workflow figure” (could be supplementary) visually showing the cohort and what data types are available for each patient/recurrent sample. There are many technologies used on these samples, in addition, there are primary & recurrent samples, it would make it easier to show the complete set up of the study.

For the WGCNA analysis, this is a more advanced bioinformatics algorithm that uses RNA expression analysis as input to model regulatory/module networks, do the authors think that enough expression data to have a reliable result? Is there any way to check the robustness of the modules discovered with WGCNA analyses?

How do these results compare with previous work, in particular the properties of the T cell cuffs with GTC cells?

What does this observation mean for treatment? In particular, what would it mean for immunotherapy? It would be great if the authors could add a few sentences to the discussion whether their findings would unlock any treatment avenues?

Reviewer #3

(Remarks to the Author)

Hijfte et al. profile a large IDHm-Astrocytoma cohort by multiplex IF, bulk RNA-seq, snRNA-seq, and spatial transcriptomics + proteomics. They show an association of gemistocytic tumor cells (GTCs) and immune cell abundance, more specifically abundance of T cells confined to the perivascular space. They use bulk and single-nucleus RNA-seq to show that GTCs are a subset of cancer cells expressing gene programs related to reactive astrogliosis and glial scarring. Using receptor-ligand analysis and IF stainings they claim to identify a network of GTCs and microglia that drive glial scarring through e.g. the SPP1/CD44 axis.

The cohort of n=76 samples, spanning all IDH-A grades is impressive. The association between GTCs and lymphocyte abundance is not completely new (Burger et al. Cancer, 1985), however analyzed more comprehensively with a focus of intratumor heterogeneity of GTC abundance. The main claims are interesting and relevant, however the analysis are at times difficult to follow and challenging to evaluate. If the authors can address these issues, I believe it is a suitable paper for Nature Communications. I appreciate the comprehensive and honest “limitations of the study” section.

Major concerns:

1. Many of the analysis is based on ROIs, such that it is difficult to evaluate how robust this finding is for the entire sample. Hence, it would be important to do the analysis also per sample and show the variability of ROI quantification within a sample. Moreover, instead of working with absolute cell numbers (which are difficult to appreciate, as the underlying area is often unclear) it would be more intuitive to work with cell densities (e.g. cells per mm²). The selection of ROIs is also unclear. In the methods the authors state that it is done for tumor cell high and low, but within these areas, where they randomly chosen to prevent any subjective bias?
2. It is often unclear what samples have been stained with which panel. An overview that depicts the multiplex staining panels and the respective cohort, should be added
3. The quality of cell segmentation (specifically challenging for the GTCs as only nuclear segmentation is used) and cell phenotyping remains completely obscure. Please add panels of representative stainings with overlapping segmentation masks (nuclear + expanded nucleus as cell boundary approximate) and a heatmap showing the staining intensity for all markers per cell type. In the methods the author state, cell segmentation was performed with the Cell detection function. This is a watershed algorithm and not a state-of-the art segmentation method (i.e. stardist, cellpose etc.). Moreover, they adjust thresholds for Cell detection and staining signal manually and per sample, which seems to introduce a subjective bias and is not reproducible. The expansion of the nucleus by 15um seems very high, especially in cell dense regions. The authors should justify this choice and show that their approach does not lead to the underestimation of TAMs in T cell cuffs.
4. The spatial relationship analyses are difficult to understand, and again a per sample quantification is missing. A better quantification and visualization should be provided. The average of minimum distances of all cell pairs in um should be added (e.g. the average distance of TAMs to the closest T cell).
5. The spatial restriction of T cells of the T cell cuffs to the perivascular space is only shown in one representative picture (Fig. 1c). This claim needs to be quantified.
6. Inferring TAM and T cell abundance from DEGs of bulk RNA-seq data seems problematic. If the authors want to do it, they should do deconvolution using e.g. CybersortX.
7. Related to Fig 5. As the authors state, the CD44 staining seems sensitive but not specific to GTCs. Hence, CD44 density as a proxy for GTC density is problematic (many cell types are CD44 positive). Given the unique staining pattern and morphology in H&E stainings, they could use a deep-learning based pixel classifier to identify the GTCs and thus, use the “real” GTC density for immune cell correlations.
8. The UMAPs of Fig. 3 are difficult to evaluate because a per sample annotation is missing (color annotation per sample). What is known in terms of GTC abundance of the seven samples used for snRNAseq? Please quantify adjacent H&E

stainings if possible. Does the expected quantities match the cell cluster 6 (=GTC) abundance? Please provide a signature for cluster 6 as this will provide a cleaner signature for CTGs than the signature from bulk RNA-seq. Moreover, the highly expressed genes in cluster 6 (shown in Figure 3c) have overlap with a mesenchymal-like program (VIM, FN1, CD44). The authors should evaluate the possibility, that GTCs transcriptionally resemble a MES-like state (similar to GBM) and maybe has been missed by previous fresh scRNA-seq studies because of a potential dissociation bias against them (very large cell bodies that seem highly interconnected). How are the markers chosen in Fig 3f? Please show that this is not cherry picking.

9. It is unclear how microglia vs BMDM were annotated in Fig. 4e. The underlying gene signatures have to be provided and show that canonical microglia marker are expressed in the microglia assigned cells (P2Y12, TMEM119, CX3CR1). It seems counterintuitive that a myeloid population expressing inflammatory genes (CCLs, IL1B, SPP1) in the context of glioma is biased towards a microglia origin. The opposite is usually the case, an enrichment in BMDM.

10. There seems to be inconsistency between the volcano plot in Fig. 3g and the heatmap in Fig S5b. Moreover, it is counterintuitive that Fibronectin and SMA are downregulated in GTC-rich ROIs as FN1 is one of the top Genes in the GTC cluster 6 and the author make the point that T cell cuff abundance is increased in GTC rich regions which comes with fibronectin in the vasculature.

11. The SPP1 staining seems highly anecdotal and needs to be quantified.

12. Github repository is empty. Multiplex images should be freely provided and not only upon request.

13. Some statements are either unclear or not fully supported by the figures they refer to:

(a) "Lymphocytes were more abundant in T cell cuffs than in stroma, independently of tumor cell numbers". How is the independence is shown?

(b) "The fraction of CD8+ T cells increased significantly together with lymphocyte clustering when compared with cuff size, whereas the inverse was seen for TAM clustering with lymphocytes". The actual results seem to be a bit more nuanced. All immune cell counts showed a significant and incremental increase associated with the GTC quantity independent of T cell related tissue phenotypes (Figure 2c; Supplementary Figure 3a,b). I'm not sure the supplementary figures show the results are independent of T-cell related tissue phenotype.

(c) Again in Figures 2d,e the results seem to more nuanced than phrased.

(d) The claim for high concordance between copy number variation estimates of snRNA-seq tumor cell clusters and genome wide DNA methylation profiles (Supplementary Figure 4d,e) should be quantified.

Minor concerns:

1. The title seems long and complicated

2. Size of ROIs for GeoMX DSP unclear. Why T cell cuffs are excluded for GeoMX DSP transcriptome?

3. Clearly state genes of signature for GTC and Module 3 TAM. Is there a gene overlap that could confound the correlation shown in Fig 5a.

4. Line 199: gene module 2 (not 4) and 5, Fig. 5a?

5. This might be out of scope, but additional stainings of the Cuffs +++ samples for a more extended immune panel to contextualize them with tertiary lymphoid structures would be desirable.

Version 1:

Reviewer comments:

Reviewer #1

(Remarks to the Author)

The requested changes have been made and I have no further suggestions.

(Remarks on code availability)

I did not notice a README file - the authors can be asked about this

Reviewer #2

(Remarks to the Author)

The authors have addressed all my comments and I recommend accepting.

(Remarks on code availability)

Reviewer #3

(Remarks to the Author)

We are satisfied by the revised manuscript and congratulate the authors for their work

(Remarks on code availability)

Comments reviewer 1:

Comment 1: Figure 1 needs more detailed and to the point analysis to demonstrate that T-cells cluster around blood vessels. They should measure distance from blood vessels in T-cells compared to other cell types in the tumor and should compare this in IDH-mut vs. IDH-wt tumors. Similarly, they should look at what % of tumor samples are considered "T-cells in perivascular space" vs. "T-cells in stroma" or "T-cells absent" in IDH-mut vs. IDH-wt gliomas.

Reply: We understand the comments regarding T cell clustering around blood vessels and thank R#1 for addressing this relevant issue. We have extended our analysis to formally test distances between T cells and endothelial cells using additional multiplex IF stainings on 15 IDH-mutant astrocytoma samples. These stainings included markers for T cells (CD3), endothelial cells (CD31) and the parenchymal basement membrane (LAMA2). Using unbiased whole-slide IF analysis we showed that T cells are located significantly closer to endothelial cells compared to other cells and that they are located behind the glia limitans. The quantifications are shown in **Figure 1f,g** and **Supplementary figure 2a-c**. The results are described in lines 89-96 of the revised manuscript:

To determine whether vessel-adjacent T cells reside in the perivascular space, we stained 15 IDHmt astrocytoma samples containing high T cell quantities for CD3, CD31 and Laminin- α 2 (LAMA2, a marker for the parenchymal basement membrane). T cells showed a significantly smaller distance to CD31+ vascular structures compared to other cells (median distance of 21 μ m and 44 μ m; Kolmogorov-Smirnov test; $D = 0.24441$, p -value $< 2.2 \times 10^{-16}$; Figure 1f, Supplementary figure 2a,b). Importantly, T cells accumulated within LAMA2 demarcated regions in all samples, indicating that T cells reside in the perivascular space behind the parenchymal basement membrane in these tumors (Figure 1e,g; Supplementary figure 2a,c).

Regarding the comments addressing the distribution of spatial phenotypes at sample level, we determined which tumor samples contain one, two or all of the specified spatial phenotypes to illustrate their distribution across our cohort (**Supplementary figure 2d**). These distributions don't significantly change between resections (chi-square test; $p = 0.073$). In our manuscript, we prioritized quantification of regions of interest (ROIs) over the percentage of tumor samples that classify according to such phenotypes due to the phenotypic variability within tumor samples (see **Figure 1c** and **Supplementary figure 3c**). Further examples of this intra-tumoral variability are depicted in **Figure 5g,h** and **Supplementary figure 8d**.

Concerning comparisons between IDHmt and IDHwt gliomas, in previous work, we have compared T cell localization between low (WHO 2016 grade II and III) and high grade gliomas (grade IV glioblastomas), where the vast majority of the grade IV tumors were IDHwt. These findings showed a stromal location of T cells in the latter tumor type, indicating that T cells more readily infiltrate these tumors compared to their (IDHmt) low grade counterparts². We added this point to our discussion to lines 279-284 and the text is added below. While it would be interesting to test differences between IDHmt and IDHwt gliomas in more detail, specifically in the context of the results presented here, performing these analyses at the level of current manuscript is outside the scope.

T cell localization does seem to differ between IDHmt and IDHwt tumors. In previous work, we showed that T cells were at a significantly farther distance from vessels in high-grade (WHO 2016 grade IV, predominantly IDHwt) compared to low grade tumors (WHO 2016 grade II/III). The reason for this difference remains to be determined but may be related to a difference in vascular structural integrity between IDHwt and IDHmt tumors.

Comment 2: Did the % of tumor samples considered "T-cells in perivascular space" change with WHO grade or with recurrence vs newly diagnosed?

Reply: As we focused our analyses on ROI level, our initial results depicted in **Figure 2d** and **Supplementary figure 3d** illustrate that the relative number of ROIs showing T cell cuffs remains similar among grades or over time.

However, we understand that comparisons at sample level aid the understanding of spatial phenotype distributions in the cohort. We therefore included figures showing the percentage of tumor samples with one or more T cell cuffs between WHO grades and initial versus recurrent resections. Both comparisons did not reveal statistically significant differences (Fisher exact test; $p = 0.936$ and $p = 1$ for between WHO grades and initial vs recurrent, respectively; **Supplementary figure 3a**). The results are described in lines 107-108 of the revised manuscript:

T cell cuffs were present in most samples and the fraction of samples containing one or more T-cell cuffs was independent of WHO grade or resection number (Supplementary figure 3a).

Comment 3: The authors need to introduce gemistocytic astrocytomas in more of an unbiased fashion with some background for those not familiar with this concept. They should perform EM analysis of the tumor samples and describe different morphologic features they see, including gemistocytic and whatever term(s) they would like to use for the non-gemistocytic tumor cells (terms like fibrillary and protoplasmic have been used in the past but are no longer recognized so it would be reasonable to just classify as gemistocytic vs. non-gemistocytic).

Reply: To introduce gemistocytic astrocytoma more clearly we added the following text at line 122-126:

GTCs are histologically distinct cells due to their large cell body with a uniformly stained eosinophilic cytoplasm and eccentrically placed nucleus. The 2021 WHO classification of central nervous system tumors (WHO 2021) assigns the label "gemistocytic astrocytoma" when >20% of malignant cells show gemistocyte morphology. Due to unclear consequences of the presence of GTCs in these tumors, attention for this tumor subtype has waned.

Additionally, we added EM microscopy images to illustrate the differences between gemistocytic and non-gemistocytic cells more clearly. In line with previous observations, these images show the buildup of fibrillary structures specifically in gemistocytic cells compared to non-gemistocytic cells (**Supplementary figure 4**). We included the relevant references in the manuscript³. The text describing our EM results has been adjusted in lines 126-128 of the revised manuscript:

Electron microscopy (EM) images of GTCs showed densely stacked filaments that were absent in other, non-gemistocytic cells, which is in concordance with other EM reports of these cells in glioma (Figure 2b, Supplementary figure 4).

Comment 4: The authors should use the cell chat algorithm on their single nuclear results to look at tumor cell communication with macrophages and T-cells in the perivascular space vs. the stroma to investigate their hypothesized mediators.

Reply: Integration of our spatial RNA profiling and snRNA-seq data specifically identified one clear association between tumor cells and immune cells; the co-localization of module 3 TAMs and gemistocytic tumor cells (GTCs) around T cell cuffs. We thus focused on communication between these two cell types in our manuscript. Along the recommendation of R#1, we have performed CellChat analysis on our snRNA-seq data, and outcomes corroborated our results, showing no specific interactions between tumor and immune cell subpopulations besides the SPP1-CD44 receptor-ligand pair between GTCs and module 3 TAMs (**Supplementary figure 7h**). Regions showing *T cell in stroma* were specifically enriched for endothelial cells and TAMs but not for any tumor cell subsets (**Supplementary figure 7d,e**). In fact, we found that most tumor cell subpopulations were spatially separated from immune cells and were enriched for gene profiles derived from *T cell absent* regions. Regarding expression profiles of *T cell cuff* ROIs, our spatial RNA experiments were designed to exclude specifically T cell cuffs (**Figure 4a**), and did not enable paring of spatial and single nucleus transcriptomics for T cells residing in the perivascular space.

Comment 5: Figure 4a spatial transcriptomics please formally define the region of what is the perivascular space and what is the T-cell cuff. We need to know exactly how close the stroma was for ROIs annotated as T cells in the perivascular space to be certain that these samples are distinct from "T-cells in the stroma" samples.

Reply: The perivascular space (also known as the Virchow–Robin space) is formally the area between the two basement membranes of the blood brain barrier. Between these two layers, T cells can accumulate. Once they accumulate to a sufficient number, we define these structures as “T cells in the perivascular space” or simply “T cell cuff”. We elaborated on the nomenclature in the manuscript at lines 88-92.

In inflamed conditions, Leukocytes can accumulate around blood vessels in the CNS perivascular space, which is delineated by the endothelial and parenchymal basement membranes. To determine whether T cells that cluster close to blood vessels reside in the perivascular space we stained 15 IDHmt astrocytoma samples containing high T cell quantities for CD3, CD31 and Laminin- α 2 (LAMA2, a marker for the parenchymal basement membrane).

For the purposes of ROI annotation in the NanoString GeoMx DSP spatial transcriptomics data, T cell cuffs were defined as dense accumulations of T cells around a vascular structure, which, as demonstrated in **Figure 1f**, reside in the perivascular space. *T cell cuff* ROIs were manually selected to specifically target tumor stroma directly adjacent to these T cell cuffs. To illustrate how closely adjacent *T cell cuff* ROIs are to the nearest T cell cuff, we have now also quantified the distance between the two. **Supplementary figure 7c** shows the near and far edge distances of T cell cuffs to *T cell cuff* ROIs, demonstrating that these ROIs specifically target tumor stroma closely adjacent to T cell cuffs. All ROIs with IF stainings and annotations are available for verification (Zenodo DOI: 10.5281/zenodo.13911761).

Comment 6: The authors used CD44 as a GTC marker but it is also an endothelial marker and endothelial cells are close spatially to GTCs. Endothelial cells can also have CNV changes in GBM so an additional marker will be needed to be confident that these are gemistocytic tumor cells.

Reply: CD44 expression is indeed not restricted to a single cell type, yet we used this marker because it was the gene most consistently associated with GTCs throughout our datasets. In our original manuscript, we corroborated CD44 stainings with additional stainings for Vimentin as a second marker for GTCs. Based on the comments by R#1, we have now included CRYAB as a third marker that is specific for GTCs. For this, we stained 15 GTC-high and 15 GTC-low IDHmt astrocytoma samples and performed whole slide image quantifications. Results show a significantly higher quantity of CRYAB+ cells in GTC-high tumors in line with earlier findings for CD44 and Vimentin (**Figure 5e**). A representative CRYAB staining is shown in **Figure 5c** alongside a HE staining of the same tumor section enabling morphological identification of GTCs. Results are described in lines 230-234 of the revised manuscript:

Since no formal marker for GTCs exists, we used CD44 and CRYAB to identify GTCs as these two markers were the most conserved across our datasets. Indeed, in our IF stainings, both CD44 and CRYAB specifically stained GTCs in both GTC-high and GTC-low tumor samples and the number of CD44+ or CRYAB+ cells were significantly higher in GTC-high tumors (Wilcoxon rank sum test; $p = 0.00666$, $p = 0.00524$, respectively; Figure 5b-e).

Comment 7: Were any canonical GBM mutations (e.g. TERT promoter, CKDN2A/B, 1p/19q, EGFR) more common in gemistocytic vs. non-gemistocytic tumor cells?

Reply: To address the question whether there were any mutation differences between GTC-high and GTC-low tumor samples, we analyzed whole exome sequencing data from the GLASS-NL patient cohort. We selected all mutations that were reliably identified (VAF > 0.05) and statistical significance was tested using a Fisher exact test. Even without correcting for multiple testing, there were no differences in frequency of canonical

glioblastoma mutations between GTC-high and GTC-low tumors. An unbiased analysis that included all identified mutations present in these samples yielded 39 mutations. None of these mutations remained statistically significant after correction for multiple testing. We included **Rebuttal figure 1** for the discretion of the reviewers. We added a comment to the manuscript at lines 155-156.

No differences in frequency of canonical mutations for glioblastomas between GTC-high and GTC-low tumors were found (data not shown).

Rebuttal figure 1; Genes that were significantly differentially mutated between GTC-low and GTC-high tumor samples. All mutations that were reliably identified in our data (VAF > 0.05) were included in the analysis and significance was tested using a Fisher exact test. Results shown are prior to multiple testing correction.

Comments reviewer 2:

Comment 1: I highly recommend the authors add a “workflow figure” (could be supplementary) visually showing the cohort and what data types are available for each patient/recurrent sample. There are many technologies used on these samples, in addition, there are primary & recurrent samples, it would make it easier to show the complete set up of the study.

Reply: We thank R#2 for requesting clarification regarding the set-up of our study, and agree that the manuscript would benefit from an overview figure showing which samples were used for which method. Following this recommendation, we have added **Figure 1a** depicting the complete study design including patient samples and techniques and added a Venn diagram (**Figure 1b**) to illustrate the sample overlap between methods. Lastly, we added an overview of all samples that were stained and analyzed for the different multiplex IF panels as **Supplementary figure 1a**.

Comment 2: For the WGCNA analysis, this is a more advanced bioinformatics algorithm that uses RNA expression analysis as input to model regulatory/module networks, do the authors think that enough expression data to have a reliable result? Is there any way to check the robustness of the modules discovered with WGCNA analyses?

Reply: To assess the reproducibility of WGCNA-identified gene modules from the current study, we randomly resampled different proportions of our dataset 100 times and analyzed resampled data using our WGCNA pipeline. We performed this analysis for 50, 60, 70, 80 and 90 percent of all the samples, after which we evaluated the extent of similarity among gene modules resulting from different sample fractions. The exercise demonstrated a reproducible and increasing overlap between gene modules when fractions of samples increased, highly suggestive that robust gene modules are formed using WGCNA (**Rebuttal figure 2a,b**). Moreover, we included a figure below from our previous work that shows a clear association between WGCNA gene modules acquired from the NanoString GeoMx DSP spatial transcriptomics data and associated ROI T cell quantity according to IF (**Rebuttal figure 2c,d**)¹. The turquoise gene module that showed the highest correlation with T cell presence and contained typical T cell marker genes with the highest importance score for T cell counts as well as the highest WGCNA module membership (a measure of connectedness for a gene in the correlation analysis of a gene module) for the turquoise module.

[REDACTED]

Rebuttal figure 2; **(a)** overlap (% of genes) between gene modules that resulted from WGCNA iterations using 50, 60, 70, 80 or 90 percent of available samples. **(b)** Violin plot of gene module overlap for 100 WGCNA iterations of randomly sampled data. First, for all comparisons, the gene overlap between the two modules that showed the greatest resemblance was calculated. Next, the mean of the gene overlap percentage was calculated and are shown in the violin plot. **(c)** Association of WGCNA gene modules with ROI T cell count. **(d)** Correlation of gene significance scores for ROI T cell count and the gene module membership score for the turquoise module. Figures c and d are adjusted from our previously published work¹.

Comment 3: How do these results compare with previous work, in particular the properties of the T cell cuffs with GTC cells?

Reply: Gemistocytic tumors are acknowledged in the WHO classification as a distinct subgroup of IDHmt astrocytomas and are defined as a tumor with more than 20% of malignant cells showing gemistocyte morphology. Earlier studies have noted T cell cuffs to occur in GTC-high tumors⁴. However, to our knowledge, large scale studies that used an integrated technological approach were never performed to quantify this observation nor to explore the underlying immune phenotype of T cell cuffs. Although some studies report the presence of SPP1 and CD44 molecules in glioma, this axis was not related to GTC-high tumors nor was it linked to T cell exclusion^{5,6}. One study characterized perivascular T cell accumulation in a glioma mouse model and was able to enhance T cell accumulation using agonistic CD40 therapy⁷. The authors of that study refer to these cuffs as tertiary lymphoid structures, a pattern that could not reliably be discerned from our data (see also response to comment 4). In general, it is worth noting that T cell cuffs have been more extensively described in other CNS pathologies, such as multiple sclerosis, which could provide further insights into characteristics of these structures in glioma and its relation to GTC and glial scarring⁸. We elaborate on this contextual information in our discussion at lines 265-270.

GTC presence is recognized as a major tissue pattern in IDHmt astrocytoma according to the WHO classification for CNS malignancies (WHO 2021), but gets little attention due to a lack of clinical relevance. The gemistocytic astrocytoma subclass has thus far been handled in a bulk fashion, disregarding tumor heterogeneity. Although some evidence already exists for a relationship between lymphocyte presence and GTCs in IDHmt astrocytoma, an in-depth examination of this immune phenotype was lacking.

Comment 4: What does this observation mean for treatment? In particular, what would it mean for immunotherapy? It would be great if the authors could add a few sentences to the discussion whether their findings would unlock any treatment avenues?

Reply: We elaborated on the implications our findings have on possible new treatment strategies in the discussion in lines 314-330.

Collectively, our findings argue that T cell exclusion in IDHmt astrocytoma is caused by specific immunological determinants. In a mouse glioma model, agonistic CD40 antibody treatment induced formation of TLSs and T cell accumulation but was unable to yield functional T cells nor enhanced responses to anti-PD1. In IDHmt astrocytoma, T cell exclusion is typically accompanied by focal presence of GTCs, and even though the focal presence of GTCs is likely transient as evidenced by our analyses of consecutive resections, the early targeting of CD44+ GTCs or nearby SPP1+ TAMs holds potential therapeutic value to break immune tolerance in this disease. For example, antibodies targeting these cells could, depending on their drug-cargo, either destruct TAMs or skew these cells into a more pro-inflammatory phenotype. Such a sensitization strategy could render IDHmt astrocytoma amenable for anti-PD1 or other immune checkpoint inhibitors. Alternatively, identification of antigen targets specifically expressed by GTCs would enable the development of adoptive T cell therapy to destruct these cells and overcome GTC-mediated T cell suppression. Currently, further research is required to intervene with glial scarring and develop therapeutics to enable T cell immunity in IDHmt astrocytoma.

Taken together, we have uncovered a cellular network between GTCs and microglia, potentially driven by reactive astrogliosis and glial scarring, that is a feature of IDHmt astrocytoma where T cells are confined to the perivascular space. Interference with this cellular network, resulting in the release of T cells, might offer a perspective for a better positioning of immune therapies to treat these tumors.

Comments reviewer 3:

Comment 1: Many of the analysis is based on ROIs, such that it is difficult to evaluate how robust this finding is for the entire sample. Hence, it would be important to do the analysis also per sample and show the variability of ROI quantification within a sample. Moreover, instead of working with absolute cell numbers (which are difficult to appreciate, as the underlying area is often unclear) it would be more intuitive to work with cell densities (e.g. cells per mm²). The selection of ROIs is also unclear. In the methods the authors state that it is done for tumor cell high and low, but within these areas, where they randomly chosen to prevent any subjective bias?

Reply:

We thank R3# for raising the incompleteness of our methodological description regarding the definition of and analyses with ROIs. Additionally, we acknowledge the relevance of analyses of the entire sample vs single ROIs. In fact, our original manuscript included whole sample analysis with regard to patient as well as cellular characteristics (**Supplementary figure 1e-i**); and with respect to the latter revealed intra- and inter-sample variability for all cell quantities (**Supplementary figure 1e**), T cell cuffs (**Supplementary figure 3c**) and GTCs (**Supplementary figure 5c**). Along R#3's recommendation, we have now extended these analyses to entire sections regarding the relationship between GTC and immune cell quantities (**Supplementary figure 5f**). These new analyses show statistically significant correlations between mean patient GTC scores with mean patient immune cell scores, but not with tumor cell scores, which confirms the patterns we have found with our ROI-based analyses. We added the following text to lines 143-145 of the revised manuscript to describe these results:

Comparison of GTC scores with cell quantities at sample level showed a correlation with immune cell, but not tumor cell quantities.

Second, concerning cell count normalization in our image analysis, we agree that using tissue area is the correct way to normalize. However, the ROIs used in this study all have the same dimensions, and normalization to tissue size would result in the same relative differences, as the normalization factor will be identical for all ROIs. We normalized all our other image analyses included in this manuscript using tissue area. We clarified this in our methods section at lines 521-522.

As all ROIs had the same dimensions, normalization to tissue area would not affect results.

Last, regarding ROI selection, we attempted to minimize the introduction of bias by using a pre-defined method; a minimum of 8 stamps of equal size were placed per sample, where possible, equally divided over tumor cell-high and tumor cell-low regions as well as over immune cell-high and immune cell-low regions. Moreover, the selection of ROIs in a given sample was performed unbiased to any clinical patient or molecular parameter. Please note that such selection of ROIs is somewhat biased towards immune cell rich regions as these were the focus of our research. Besides the above selection method, and appreciating the risk of bias, we extensively validated our ROI-based results, including whole slide image analysis and bulk RNA sequencing (see **Figure 5**). We extended our methods to more clearly outline the ROI selection procedure in lines 504-507 of the adjusted manuscript.

Following whole slide scans using VECTRA 3.0 (Akoya Biosciences), at least eight ROIs (size: 670 × 502 μm²; pixel size: 0.5 × 0.5 μm²) were equally divided over tumor cell-high and tumor cell-low regions as well as over immune cell-high and immune cell-low regions. The selection of ROIs in a given sample was performed unbiased to any clinical patient or molecular parameter.

Comment 2: It is often unclear what samples have been stained with which panel. An overview that depicts the multiplex staining panels and the respective cohort, should be added.

Reply: In line with our response to R#2 comment 1, we have added an overview of which samples have been stained and analyzed in **Supplementary figure 1a**. More broadly, to make the complete study setup more comprehensible, we provide a study overview in **Figure 1a,b**.

Comment 3: The quality of cell segmentation (specifically challenging for the GTCs as only nuclear segmentation is used) and cell phenotyping remains completely obscure. Please add panels of representative stainings with

overlapping segmentation masks (nuclear + expanded nucleus as cell boundary approximate) and a heatmap showing the staining intensity for all markers per cell type. In the methods the author state, cell segmentation was performed with the Cell detection function. This is a watershed algorithm and not a state-of-the art segmentation method (i.e. stardist, cellpose etc.). Moreover, they adjust thresholds for Cell detection and staining signal manually and per sample, which seems to introduce a subjective bias and is not reproducible.

Reply: In our revised manuscript, we have now added representative images of all stainings along with their segmentation masks (**Supplementary figure 1b**, **Supplementary figure 2a** and **Supplementary figure 8a**). In addition, we have included heatmaps depicting signal intensities per staining (**Supplementary figure 1d** and **Supplementary figure 8b**). Next to this, we now reanalyzed all whole slide samples using the StarDist algorithm for cell segmentation. Finally, we have used standardized scripts for all stainings and specified which settings were used in our methods section in lines 533-536 of the revised text:

Within tissue regions, cell detection was performed using the StarDist extension in QuPath. The standard dsb2018_heavy_augment model was used. NormalizePercentiles were set to 0.1 and 99.9, tileSize was set to 1500, Probability threshold was set to 0.5, cellExpansion was set to 5 μ m, and cellConstrainScale was set to 1.5. Methods were identical for all whole slide image analyses.

With respect to the point raised about manually thresholding signals for each sample, we argue that it this, at least in our experience, is inherent to working with FFPE tumor materials. Nevertheless, to limit any potential bias, thresholding has been performed by two independent observers and has always been blinded towards clinical or molecular information of any given sample. Importantly, and enabling reproduction of our findings, all thresholds used are listed in our **Supplementary table 7** and all whole slide multiplex images are available on Zenodo (CRYAB/HE: 10.5281/zenodo.13918123; CD31, CD3, LAMA2: 10.5281/zenodo.13918377; CD44, CD3, SPP1, TNC and CD68, IDH1-R132H, CD3, CD31: 10.5281/zenodo.13911899).

Comment 4: The spatial relationship analyses are difficult to understand, and again a per sample quantification is missing. A better quantification and visualization should be provided. The average of minimum distances of all cell pairs in μ m should be added (e.g. the average distance of TAMs to the closest T cell).

Reply: To address this question, we have now added per sample cell quantities of cell types as well as nearest neighbor (NN) analysis for our whole slide image analysis using two multiplex IF stainings of consecutive slides (CD3/CD44; CD3/CD68; **Figure 5d,f**). Cell quantities show expected patterns, with a significantly higher CD44+ cell number in GTC-high tumors (Wilcoxon rank sum test; $p = 0.00666$). NN analysis showed both T cells as well as GTCs cluster together independent of GTC status. T cells did not cluster together with either GTCs or TAMs, which is in line with the observed perivascular T cell clustering according to our ROI-based analysis. The only statistically significant difference between GTC-high and GTC-low samples was the clustering among GTCs, which appeared more frequently in GTC-low samples. Although counterintuitive, this is likely due to a less ubiquitous presence of GTCs in GTC-low samples. Last, to further clarify the spatial analyses we have performed, the CD44 kernel density analysis using whole slide images is now illustrated in **Figure 5g**. These new results are described in lines 230-243 of the revised text:

Since no formal marker for GTCs exists, we used CD44 and CRYAB to identify GTCs as these were the two most conserved markers across our datasets. Indeed, in our IF stainings, both CD44 and CRYAB showed specific signal for GTCs in both GTC-high and GTC-low tumor samples and numbers of CD44 or CRYAB+ cells were significantly enriched in GTC-high tumors (Wilcoxon rank sum test; $p = 0.00666$, $p = 0.00524$, respectively; Figure 5b-e). Whole slide nearest neighbor (NN) analysis showed consistent clustering within a cell type (GTCs, TAMs and T cells) independent of GTC abundance, but not between cell types (GTCs with T cells or TAMs with T cells) in both GTC high and low tumors (Figure 5f). Next, we sought to assess how local CD44 densities compared to that of T cells or TAMs. To this end, we have calculated normalized local CD44 kernel densities in whole slide images (Figure 5g). To test which cell types increased alongside local CD44 density, all cell locations were assigned to a CD44 density value in each sample. Cell CD44 density values from all samples were pooled and separated into 20 bins (Figure 5g). High CD44 density bins were almost uniquely occupied by GTC-high samples and high T cell fractions exclusively increased in CD44 dense regions across GTC-high samples (Figure 5h-j).

Comment 5: The spatial restriction of T cells of the T cell cuffs to the perivascular space is only shown in one representative picture (Fig. 1c). This claim needs to be quantified.

Reply: We have now included spatial analysis of 15 GTC-high tumors stained for CD31, CD3 and LAMA2 (see also our response to R#1, comment 1). These markers enable demarcation of T cells (CD3) that reside in the perivascular space, i.e., the space between endothelial cells (CD31) and the parenchymal basement membrane (LAMA2). Cell detection was performed with the StarDist algorithm and both CD31 and LAMA2-positive areas were detected by training a pixel classifier in QuPath (see methods). An overview of the workflow is shown in **Supplementary figure 2a**. T cell cuffs were defined as three or more CD3+ cells encircled by LAMA2+ regions, and were found in all 15 samples. Cuffs displayed a large range in T cell quantity and the number of cuffs per tissue slice was also highly variable (**Supplementary figure 2c**). Importantly, our analysis showed a statistically significant higher T cell quantity in LAMA2 demarcated regions versus stroma, implying that T cells are indeed confined to the perivascular space (**Figure 1g**). Results are explained in lines 94-96 in the revised manuscript.

Importantly, T cells accumulated within LAMA2 demarcated regions in all samples, indicating that T cells reside in the perivascular space behind the parenchymal basement membrane in these tumors (Figure 1e,g; Supplementary figure 2a,c).

Comment 6: Inferring TAM and T cell abundance from DEGs of bulk RNA-seq data seems problematic. If the authors want to do it, they should do deconvolution using e.g. CyberSortX.

Reply: The analyses shown in **Figure 2f,g** illustrate that *genes* and *pathways* associated with immune cell infiltration and activation are upregulated in GTC-high vs GTC-low tumors. Although these data are in line with our IF analysis, and suggest an increase in immune cell numbers in GTC-high vs GTC-low tumors, we have been cautious throughout our manuscript to make inferences on immune cell quantities based on our bulk RNA sequencing data. All quantitative analyses relied solely on IF stainings.

However, and in response to R#3, we have now also performed CyberSortX and Quantiseq analyses to infer immune cell quantities from our bulk RNA-seq data, using widely used public reference datasets⁹. The results from both deconvolution tools differed greatly and did not recapitulate cell quantities from our IF results, nor patterns that could be expected based on literature (**Rebuttal figure 3**)¹⁰. For example, both methods underestimated the frequency of myeloid cells in IDHmt astrocytoma. Furthermore, CyberSortX overestimated tumor purity compared to our estimates of tumor purity according to shallow whole genome sequencing (ACE cellularity) and DNA methylation (RF purity Absolute, see <https://doi.org/10.1101/2024.03.05.583306>)¹¹. However, published data were in line with the IF quantifications of our datasets¹². These results show that cell deconvolution from bulk RNA data can be precarious and, for reasons currently unknown, are not reproducible. Therefore, we used a relatively straightforward result in our manuscript that only relies on increased expression of marker genes associated with immune cell types to illustrate that the expression patterns are in line with increased presence of immune cells, as shown in our ROI image analysis. We have included the figures here for the reviewers discretion (**Rebuttal figure 3**).

Figure 3; **(a)** Bar plot of cell type deconvolution from CybersortX for all samples according to ground truth published in Varn et al (2020). **(b)** Boxplot comparisons of relative cell abundance according to CyberSortX between GTC-high and GTC-low tumors. Statistical significance was tested with the Wilcoxon rank sum test. **(c)** Bar plot of cell type deconvolution according to Quantiseq for all samples according to the standard reference. **(d)** Boxplot comparisons of relative cell abundance according to Quantiseq between GTC-high and GTC-low sample groups according to our study. Statistical significance was tested with the Wilcoxon rank sum test. Data in b and d show mean \pm SD, p-value: * <0.05 ; ** <0.01 ; *** <0.001 ; **** <0.0001 .

Comment 7: Related to Fig 5. As the authors state, the CD44 staining seems sensitive but not specific to GTCs. Hence, CD44 density as a proxy for GTC density is problematic (many cell types are CD44 positive). Given the unique staining pattern and morphology in H&E stainings, they could use a deep-learning based pixel classifier to identify the GTCs and thus, use the “real” GTC density for immune cell correlations.

Reply: HE stainings could be used for automated identification of GTCs and we acknowledge that CD44 is ubiquitously expressed, as highlighted in our *Limitations to the study* section. However, the use of a protein marker offers several advantages; due to the consistent and comparatively remarkably high expression of CD44 in GTCs (**Rebuttal figure 4**) in all our datasets, as well as the implication of its receptor function for SPP1 secreted by module 3 TAMs, CD44 staining serves both as a marker for GTCs and as an indicator for important pathways in the TME organization of IDH-mutant astrocytoma. Additionally, protein markers offer the possibility for a direct spatial comparison with other cell types (e.g. T cells) in IF stainings. Lastly, the IF stainings with GTC markers served as validation of our snRNA-seq GTC profiles.

Still, CD44 as a single marker to confidently identify GTCs is contentious. We therefore now included an additional marker for GTCs (CRYAB) to substantiate the identification of GTCs with protein markers, and assessed this marker in 15 GTC-high and 15 GTC-low samples. See also our responses to R#1, comment 6 for details. Comparisons of CRYAB IF and HE stainings of the same tissue slide showed that CRYAB specifically stains GTCs in our samples (**Figure 5c**). The number of CRYAB+ cells were significantly higher in GTC-high versus GTC-low tumors (**Figure 5e**). The results are described in lines 230-234 of the revised manuscript.

Since no formal marker for GTCs exists, we used CD44 and CRYAB to identify GTCs as these two markers were the most conserved across our datasets. Indeed, in our IF stainings, both CD44 and CRYAB specifically stained GTCs in both GTC-high and GTC-low tumor samples and the number of CD44+ or CRYAB+ cells were significantly higher in GTC-high tumors (Wilcoxon rank sum test; $p = 0.00666$, $p = 0.00524$, respectively; Figure 5b-e).

Rebuttal figure 4; UMAP representations of all nuclei integrated from 7 tumors showing cell type annotations (left) and CD44 expression (right).

Comment 8: The UMAPs of Fig. 3 are difficult to evaluate because a per sample annotation is missing (color annotation per sample). What is known in terms of GTC abundance of the seven samples used for snRNAseq? Please quantify adjacent H&E stainings if possible. Does the expected quantities match the cell cluster 6 (=GTC) abundance? Please provide a signature for cluster 6 as this will provide a cleaner signature for CTGs than the signature from bulk RNA-seq. Moreover, the highly expressed genes in cluster 6 (shown in Figure 3c) have overlap with a mesenchymal-like program (VIM, FN1, CD44). The authors should evaluate the possibility, that GTCs transcriptionally resemble a MES-like state (similar to GBM) and maybe has been missed by previous fresh scRNA-seq studies because of a potential dissociation bias against them (very large cell bodies that seem highly interconnected). How are the markers chosen in Fig 3f? Please show that this is not cherry picking.

Reply: In response to the above comment, we would like to put forward the following points.

- We have added all ‘per sample annotations’ to **Supplementary figure 5i**.
- For all snRNA-seq tumor samples, we evaluated HE stainings from both cryopreserved and FFPE preserved tissue. All stainings showed a degree of GTC presence (**Supplementary figure 6d**). However, due to our

sample processing (we process tissue blocks instead of tissue sections for snRNA-seq), we cannot accurately predict how many GTCs will be present in the snRNA-seq data from the processed cryopreserved tissue, as their presence is often highly focal, as is evident from **Supplementary figure 5c**, that shows variability in GTC scores between ROIs within an individual tumor samples.

- We agree that in the UMAP, there is clear overlap between Seurat cluster 6 and the GTC cluster that is identified using the enrichment score for bulk RNA seq DE genes. We included a figure to highlight the overlap of the differentially expressed genes between the two cell populations which indeed shows a large overlap (**Rebuttal figure 5a**). We now added differentially expressed genes for all Seurat cell clusters to **Supplementary table 12**. Please note that only a fraction of the cells that belong to cluster 6 are allocated to the GTC cluster, and that nuclei scoring high for the GTC signature are also present in clusters 10, 0, 2 and 4 (**Rebuttal figure 5b,c**). The absence of a pure single cell GTC cluster supports the validity of comparing the bulk signature derived from GTC high samples with individual nuclei directly.
- We acknowledge the resemblance of the mesenchymal tumor cell state in GBM and the GTC state identified here and thank R#3 for making this comment. In fact, recent studies indicate that the mesenchymal cell state in glioblastomas are involved in scarring mechanisms, similar to our findings in IDHmt astrocytoma, highlighting similar signaling pathways in both diseases linked to a glial-scarring response^{6,13}. We adapted our discussion in lines 310-313 of the revised manuscript:

Of note, GTC signature genes show overlap with the mesenchymal state in glioblastoma. Recently, MES like cells have been linked to glial-like wound response⁶. Possibly, glial scarring mechanisms identified in here are more generally present in primary brain tumors.

- There is a well-known dissociation bias in both single cell and single nucleus RNA sequencing for various cell types including immune cells, neurons and astrocytes. GTCs might be affected by this bias as well due to their morphology. Even though this selection bias may have contributed to the identification of GTCs in our data, we do want to highlight that we took a supervised approach to identify GTCs based in HE and multiplex IF stainings, which is different from most other approaches used to identify malignant cell states in these types of data. We added the following text to our discussion to lines 261-265 to highlight these possibilities.

We were able to identify the GTC cell state for the first time using a bulk RNA-seq informed analysis of snRNA-seq data. Other single cell studies that evaluated IDHmt astrocytoma at single cell level did not discern a specific GTC cell state, which could be due to the targeted approach on the GTC cell state used in this study. Alternatively, a negative selection bias against large interconnected cells (like GTCs) in dissociation protocols used in most studies could limit their ability to detect GTCs.

- The genes shown in **Figure 3f** are derived from reported reactive astrocyte-specific gene lists (**Supplementary table 11**). The dot plot shows overlapping genes between the top 2,000 most variable genes from the integrated snRNA-seq tumor cell dataset and from the reactive astrocyte marker gene list, as is standard in these types of analyses. No further selections have been made to this gene list. We have updated the method section regarding gene lists for reactive astrocytes. This is described in method section in lines 694-697:

Markers for reactive astrocytes were collected from literature without any further filtering or selection (Table S11). For enrichment calculations, overlapping genes between marker gene sets and the top 2,000 most variable genes from the respective snRNA-seq datasets were used.

Rebuttal figure 5; **(a)** Overlap of marker genes from Seurat cell cluster 6 and the GTC cluster identified using the bulk RNA-seq GTC signature. **(b)** Density plot of the GTC enrichment scores in tumor cells. Vertical line shows cutoff value for GTCs. **(c)** Violin plot showing the GTC enrichment scores for cells in the different Seurat cell clusters. Horizontal line shows the cutoff value for GTCs.

Comment 9: It is unclear how microglia vs BMDM were annotated in Fig. 4e. The underlying gene signatures have to be provided and show that canonical microglia marker are expressed in the microglia assigned cells (P2Y12, TMEM119, CX3CR1). It seems counterintuitive that a myeloid population expressing inflammatory genes (CCLs, IL1B, SPP1) in the context of glioma is biased towards a microglia origin. The opposite is usually the case, an enrichment in BMDM.

Reply: All marker genes that were used have been made available in **Supplementary table 7**. We show the expression of the canonical microglia markers mentioned by R#3 in **Rebuttal figure 6** for reference. In our work, we were very careful in assigning the origin of TAMs based on snRNA-seq data as BMDMs and microglia transcriptomically tend to form a spectrum in which it is difficult to set a clear cutoff¹⁴. Instead, we used a combination of enrichment scores for BMDMs and microglia gene signatures from literature to indicate whether cells in the TAM subpopulation tended more towards a BMDM or microglia expression profile (hence the scale indicated by the color gradient in **Figure 4e**)¹⁴. This showed that within the spectrum, more cells were enriched for the microglia gene signature. We never assigned TAMs in our snRNA-seq data to a BMDM or microglial origin. In **Figure 4f** we specifically tested the TAMs that were positive for gene module 3 and showed that module 3 TAMs are enriched for the microglial gene signature.

It is certainly the case in GBM that BMDMs are in the majority. For IDHmt astrocytoma however, we and others find that the majority of recruited TAMs are of microglial origin¹⁰. The expression of inflammatory genes by microglia has also been reported in glioma, and is more generally seen in glial scarring processes in the CNS^{5,15}. Given that we find that glial scarring signatures upregulated in gemistocytic tumor cells, co-localizing microglia that locally express accompanying inflammatory “reactive” programs is, in our view, plausible. We have now noted this in our discussion at lines 306-308:

Notably, our TAM expression profiles were predominantly enriched for microglia markers, which is in line with reports showing that the majority of recruited TAMs are of microglial origin in IDHmt astrocytoma.

Rebuttal figure 6; UMAP representation of TAMs showing expression of three canonical microglia markers.

Comment 10: There seems to be inconsistency between the volcano plot in Fig. 3g and the heatmap in Fig S5b. Moreover, it is counterintuitive that Fibronectin and SMA are downregulated in GTC-rich ROIs as FN1 is one of the top Genes in the GTC cluster 6 and the author make the point that T cell cuff abundance is increased in GTC rich regions which comes with fibronectin in the vasculature.

Reply: To prevent confusion between the two visualizations and to clarify the analysis of the protein data, we have extended our methods section. The differential protein expression test was performed by using GTC-low and GTC-high ROI groups (**Figure 3g**), whereas the heatmap shows unbiased clustering of ROIs based on all protein data, in which the sample annotations show that clustering was largely determined by T cell quantity and to a lesser extent by GTC quantity (**Supplementary figure 7b**). We have validated this analysis and confirm that our tests were correct. The expanded text of our method section can be found at lines 661-664.

Normalized counts of the NanoString data were rounded and sizeFactors were set to 1 to perform differential gene expression tests based on ROI GTC scores. Separately, unbiased clustering of NanoString GeoMx DSP protein data was performed using the ComplexHeatmap R package.

The discrepancy between bulk RNA expression and spatial protein presence is understandably confusing. Our protein data was collected from 72 ROIs throughout 12 samples. We find that the data from the ROIs separated based on local tissue phenotypes rather than individual samples. The presence of local protein quantities from relatively small ROIs is difficult to compare to bulk RNA profiles, as they represent biology differently in various aspects. Additionally, the strategy for ROI placement differed between spatial RNA and protein experiments, where *T cell cuff* ROIs from our protein data specifically selected the T cell cuffs (including the entire cuff) whereas tumor stroma surrounding T cell cuffs (excluding the cuff) was selected for our RNA data (**Supplementary figure 7a**). As a result, T cell cuff areas used for protein data were underrepresented regarding GTCs, and overrepresented regarding vessels. This nuance in the data could result in increased Fibronectin and SMA protein levels in GTC-low areas. We highlight these differences and warrant caution for direct comparisons of the data modalities in the limitations of the study section in lines 355-361:

Nuances between the different data formats used in this manuscript can make comparisons between them difficult and should be performed with caution. First, ROI selection for spatial RNA and protein assays are not identical; ROIs annotated as T cells in perivascular space in the protein assay specifically select T cells, whereas the adjacent tumor stroma was selected in the RNA assay. Second, bulk and spatial expression profiles should be compared with some restraint, as the ROIs represent local biological processes within one tumor, and bulk measurements represent global biological patterns that might not represent local expression patterns.

Comment 11: The SPP1 staining seems highly anecdotal and needs to be quantified.

Reply: R#3's comment is in line with what was mentioned in the section limitations. Indeed, it proved difficult to properly quantify SPP1 levels. Even though we have clear examples of the presence of SPP1 in GTC-high tumors, the overall staining and analysis was not robust, most likely because SPP1 is a secreted factor that can get diluted in the tissue. In general, staining and quantification of secreted factors in tissues provides a technical challenge. We therefore show representative images of SPP1 and IL1 β staining in these tumors. Our *limitations to the study* section is extended with lines 352-354 to highlight this technical challenge:

Since SPP1 is a soluble factor, it proved difficult to quantify its presence in tissues. We therefore used single marker chromogenic IHC stainings for SPP1 instead of multiplex IF stainings to confirm the presence of SPP1 specifically in GTC-high tumors.

Comment 12: Github repository is empty. Multiplex images should be freely provided and not only upon request.

Reply: All imaging data as well as the sequencing data have been made available on Zenodo (CRYAB/HE: 10.5281/zenodo.13918123; CD31, CD3, LAMA2: 10.5281/zenodo.13918377; CD44, CD3, SPP1, TNC and CD68, IDH1-R132H, CD3, CD31: 10.5281/zenodo.13911899; NanoString GeoMx spatial transcriptomics ROIs: DOI: 10.5281/zenodo.13911761; NanoString GeoMx spatial proteomics ROIs: DOI: 10.5281/zenodo.13918032) and the corresponding R scripts have been deposited on GitHub (https://github.com/LevivanHijfte/GTC_paper).

Comment 13: Some statements are either unclear or not fully supported by the figures they refer to:

(a) "Lymphocytes were more abundant in T cell cuffs than in stroma, independently of tumor cell numbers". How is the independence shown?

(b) "The fraction of CD8+ T cells increased significantly together with lymphocyte clustering when compared with cuff size, whereas the inverse was seen for TAM clustering with lymphocytes". The actual results seem to be a bit more nuanced. All immune cell counts showed a significant and incremental increase associated with the GTC

quantity independent of T cell related tissue phenotypes (Figure 2c; Supplementary Figure 3a,b). I'm not sure the supplementary figures show the results are independent of T-cell related tissue phenotype.

(c) Again in Figures 2d,e the results seem to more nuanced than phrased.

(d) The claim for high concordance between copy number variation estimates of snRNA-seq tumor cell clusters and genome wide DNA methylation profiles (Supplementary Figure 4d,e) should be quantified.

Reply: Our response to the above 4 points is as follows.

a. The statement that statistically significant differences in immune cell abundances between spatial T cell phenotypes is independent of tumor cell quantity refers to **Figure 1h** and is based on the lack of differences in tumor cell quantities between the T cell phenotypes. To further address this point, we have now tested whether immune cell abundances remain different when corrected for ROI tumor cell quantity (**Supplementary figure 2e**). Our analysis showed that, when corrected for tumor cell count, immune cell abundances remained similarly significantly different.

b. The sentence that is referred to describes a NN analysis in which lymphocyte subsets show increased clustering depending on T cell cuff size score, and decreased clustering of TAMs. In the same sentence, we also referred to the significant increase of CD8+ T cell fraction in all T cells in ROIs depending on ROI T cell cuff size score. To prevent any confusion, we have adjusted lines 111-114 as follows:

Lymphocyte subset clustering increased significantly together with cuff size, whereas the inverse was seen for TAM clustering with lymphocytes (Supplementary figure 3f). Remarkably, within the T cell population, the fraction of CD8+ T cells increased significantly with an increase in cuff size (Supplementary figure 3g).

c. Lines 138-140 of the text have been revised to reflect a the more gradual nature of our results:

T cell cuff ROIs showed a significantly higher abundance of GTCs compared to other T cell related spatial phenotypes, even though GTCs were present within all spatial T cell phenotypes. The quantity and size of T cell cuffs positively correlated with GTC quantity (chi-square test; $p < 0.01$ for all tests between the three tissue phenotypes, Figure 2d; chi-square test; $p < 2.2 \times 10^{-16}$; Figure 2e).

d. To address this question, we have now assessed the concordance between copy number changes quantified using bulk methylation and snRNA-seq data. These data show a statistically significant positive correlation between the two methods (**Supplementary figure 6g**). Of note, the bulk methylation data identified more CNVs than snRNAseq data, likely due to the better resolution.

Minor concerns:

1. The title seems long and complicated.

Reply: Title will be adjusted to: *Gemistocytic tumor cells programmed for glial scarring characterize T cell confinement in IDH-mutant astrocytoma*

2. Size of ROIs for GeoMX DSP unclear. Why T cell cuffs are excluded for GeoMX DSP transcriptome?

Reply: All ROI sizes for the NanoString GeoMX DSP are included in the **Supplementary table 5**. T cell cuffs were excluded from ROI selection because our study specifically focused on cells in the tumor stroma directly adjacent to T cell cuffs and that potentially limit T cell extravasation and infiltration.

3. Clearly state genes of signature for GTC and Module 3 TAM. Is there a gene overlap that could confound the correlation shown in Fig 5a.

Reply: The gene signatures for GTCs and Module 3 TAMs are included in **Supplementary table 6**. Regarding the analysis shown in **Figure 5a**, we removed any duplicate markers from the gene lists in the analysis to ensure these genes did not confound results.

4. Line 199: gene module 2 (not 4) and 5, Fig. 5a?

Reply: This mistake has been corrected.

5. This might be out of scope, but additional stainings of the Cuffs +++ samples for a more extended immune panel to contextualize them with tertiary lymphoid structures would be desirable.

Reply: We underscore the relevance of additional immune phenotyping of tumors with a high density of T cell cuffs. Importantly, our data do not corroborate that T cell cuffs represent classical TLS as we see no apparent organization in the leukocyte accumulation (**Figure 1d**). However, our data do associate T cell cuffs with the existence of a novel cellular network of GTCs and reactive microglial cells, including ligand-receptor pairs that enable communication between the two cell types. To further complement these phenotypes, and ultimately perform intervention experiments to enhance T cell immunity in IDHmt astrocytoma, further research is recommended in our new discussion section and the current lack of additional phenotyping is mentioned in our limitations.

References

1. van Hijfte, L., Geurts, M., Vallentgoed, W.R., Eilers, P.H.C., Sillevs Smitt, P.A.E., Debets, R., and French, P.J. (2023). Alternative normalization and analysis pipeline to address systematic bias in NanoString GeoMx Digital Spatial Profiling data. *iScience* 26, 105760. S2589-0042(22)02033-8 [pii] 105760 [pii] 10.1016/j.isci.2022.105760.
2. Weenink, B., Draaisma, K., Ooi, H.Z., Kros, J.M., Sillevs Smitt, P.A.E., Debets, R., and French, P.J. (2019). Low-grade glioma harbors few CD8 T cells, which is accompanied by decreased expression of chemo-attractants, not immunogenic antigens. *Sci Rep* 9, 14643. 10.1038/s41598-019-51063-6 10.1038/s41598-019-51063-6 [pii].
3. Kros, J.M., Stefanko, S.Z., de Jong, A.A.W., van Vroonhoven, C.C.J., van der Heul, R.O., and van der Kwast, T.H. (1991). Ultrastructural and immunohistochemical segregation of gemistocytic subsets. *Human pathology* 22, 33-40.
4. Krouwer, H.G., Davis, R.L., Silver, P., and Prados, M. (1991). Gemistocytic astrocytomas: a reappraisal. *J Neurosurg* 74, 399-406. 10.3171/jns.1991.74.3.0399.
5. He, C., Sheng, L., Pan, D., Jiang, S., Ding, L., Ma, X., Liu, Y., and Jia, D. (2021). Single-Cell Transcriptomic Analysis Revealed a Critical Role of SPP1/CD44-Mediated Crosstalk Between Macrophages and Cancer Cells in Glioma. *Front Cell Dev Biol* 9, 779319. 779319 [pii] 10.3389/fcell.2021.779319.
6. Mossi Albiach, A., Janusauskas, J., Kapustová, I., Kvedaraite, E., Codeluppi, S., Munting, J.B., Borm, L.E., Kjaer Jacobsen, J., Shamikh, A., Persson, O., and Linnarsson, S. (2023). Glioblastoma is spatially organized by neurodevelopmental programs and a glial-like wound healing response. *bioRxiv*, 2023.2009.2001.555882. 10.1101/2023.09.01.555882.
7. van Hooren, L., Vaccaro, A., Ramachandran, M., Vazaios, K., Libard, S., van de Walle, T., Georganaki, M., Huang, H., Pietila, I., Lau, J., et al. (2021). Agonistic CD40 therapy induces tertiary lymphoid structures but impairs responses to checkpoint blockade in glioma. *Nat Commun* 12, 4127. 10.1038/s41467-021-24347-7 [pii] 24347 [pii] 10.1038/s41467-021-24347-7.
8. Smolders, J., Fransen, N.L., Hsiao, C.C., Hamann, J., and Huitinga, I. (2020). Perivascular tissue resident memory T cells as therapeutic target in multiple sclerosis. *Expert Rev Neurother* 20, 835-848. 10.1080/14737175.2020.1776609.
9. Varn, F.S., Johnson, K.C., Martinek, J., Huse, J.T., Nasrallah, M.P., Wesseling, P., Cooper, L.A.D., Malta, T.M., Wade, T.E., Sabedot, T.S., et al. (2022). Glioma progression is shaped by genetic evolution and microenvironment interactions. *Cell* 185, 2184-2199 e2116. S0092-8674(22)00536-0 [pii] 10.1016/j.cell.2022.04.038.
10. Klemm, F., Maas, R.R., Bowman, R.L., Kornete, M., Soukup, K., Nassiri, S., Brouland, J.P., Iacobuzio-Donahue, C.A., Brennan, C., Tabar, V., et al. (2020). Interrogation of the Microenvironmental Landscape in Brain Tumors Reveals Disease-Specific Alterations of Immune Cells. *Cell* 181, 1643-1660 e1617. S0092-8674(20)30569-9 [pii] 10.1016/j.cell.2020.05.007.
11. Wies, R.V., Youri, H., Karin, A.v.G., Levi van, H., Erik van, D., Mathilde, C.M.K., Johanna, M.N., Kaspar, D., Ivonne, M., Wendy, W.J.d.L., et al. (2024). Evolutionary trajectories of IDH-mutant astrocytoma identify molecular grading markers related to cell cycling. *bioRxiv*, 2024.2003.2005.583306. 10.1101/2024.03.05.583306.
12. Friebel, E., Kapolou, K., Unger, S., Nunez, N.G., Utz, S., Rushing, E.J., Regli, L., Weller, M., Greter, M., Tugues, S., et al. (2020). Single-Cell Mapping of Human Brain Cancer Reveals Tumor-Specific Instruction of Tissue-Invasive Leukocytes. *Cell* 181, 1626-1642 e1620. S0092-8674(20)30561-4 [pii] 10.1016/j.cell.2020.04.055.
13. Ruiz-Moreno, C., Salas, S.M., Samuelsson, E., Brandner, S., Kranendonk, M.E.G., Nilsson, M., and Stunnenberg, H.G. (2022). Harmonized single-cell landscape, intercellular crosstalk and tumor architecture of glioblastoma. *bioRxiv*, 2022.2008.2027.505439. 10.1101/2022.08.27.505439.
14. Venteicher, A.S., Tirosh, I., Hebert, C., Yizhak, K., Neftel, C., Filbin, M.G., Hovestadt, V., Escalante, L.E., Shaw, M.L., Rodman, C., et al. (2017). Decoupling genetics, lineages, and microenvironment in IDH-mutant gliomas by single-cell RNA-seq. *Science* 355, 355/6332/eaai8478 [pii] 10.1126/science.aai8478.
15. Sofroniew, M.V. (2014). Astrogliosis. *Cold Spring Harb Perspect Biol* 7, a020420. cshperspect.a020420 [pii] a020420 [pii] 10.1101/cshperspect.a020420.

Reviewers comments:

Reviewer #1:

The requested changes have been made and I have no further suggestions.

Reply: We thank referee #1 for this positive assessment.

Reviewer #1 (Remarks on code availability):

I did not notice a README file - the authors can be asked about this

Reply: A README file has been added.

Reviewer #2 (Remarks to the Author):

The authors have addressed all my comments and I recommend accepting.

Reply: We thank referee #2 for this positive assessment.

Reviewer #3 (Remarks to the Author):

We are satisfied by the revised manuscript and congratulate the authors for their work.

Reply: We thank referee #3 for this positive assessment.